# Better Estimation of the Kullback–Leibler Divergence Between Language Models

**Afra Amini**    **Tim Vieira**    **Ryan Cotterell**
ETH Zürich
{afra.amini, ryan.cotterell}@inf.ethz.ch
tim.f.vieira@gmail.com

## Abstract

Estimating the Kullback–Leibler (KL) divergence between language models has many applications, e.g., reinforcement learning from human feedback (RLHF), interpretability, and knowledge distillation. However, computing the exact KL divergence between two arbitrary language models is intractable. Thus, practitioners often resort to sampling-based estimators. While it is easy to fashion a simple Monte Carlo (MC) estimator that provides an unbiased estimate of the KL divergence between language models, this estimator notoriously suffers from high variance and can even result in a negative estimate of the KL divergence, a non-negative quantity. In this paper, we introduce a Rao–Blackwellized estimator that is unbiased and provably has variance less than or equal to that of the standard Monte Carlo estimator. In an empirical study on sentiment-controlled fine-tuning, we show that our estimator provides more stable KL estimates and reduces variance substantially. Additionally, we derive an analogous Rao–Blackwellized estimator of the gradient of the KL divergence, which leads to more stable training and produces models that more frequently appear on the Pareto frontier of reward vs. KL compared to the ones trained with the MC estimator of the gradient.

## 1 Introduction

The Kullback–Leibler [KL; 19] divergence is a statistical divergence that quantifies how one probability distribution differs from another. Measuring the KL divergence between probability distributions is a well-established problem that has been studied extensively in the statistics literature [7, 12, *inter alia*]. In some special cases, e.g., in the case that we wish to measure the KL divergence between two Gaussian measures, the KL divergence has an analytical solution. However, in the general case, exact computation of the KL divergence is not analytically tractable or approximable with an efficient algorithm [14]. This paper treats the case of computing the KL divergence between two language models (LMs), a fundamental task in natural language processing with numerous practical applications.

The KL divergence plays a central role across multiple applications. In reinforcement learning from human feedback [RLHF; 6, 26, 35], it is used as a regularization term to constrain the fine-tuned model from drifting too far from a reference model, preserving fluency and preventing reward over-optimization. In interpretability research, KL divergence quantifies how a specific prompt shifts the model distribution by comparing the model's distributions before and after controlled interventions [9, 27, 36]. As an evaluation metric, KL divergence is used to assess how well language models approximate target distributions [4, 37]. In knowledge distillation, KL divergence is minimized to align a student model with a teacher model [1].

39th Conference on Neural Information Processing Systems (NeurIPS 2025).

The above applications demonstrate that measuring the KL divergence between two language models is useful and widespread. However, in the case of neural language models, it is far from straightforward. It is easy to see why: Given an alphabet of symbols $\Sigma$ and two language models $p$ and $q$, distributions over $\Sigma^*$, the **KL divergence** is given by the following expression:[1,2]

$$\text{KL}(p \parallel q) \stackrel{\text{def}}{=} \sum_{\boldsymbol{y} \in \Sigma^*} p(\boldsymbol{y}) \log \frac{p(\boldsymbol{y})}{q(\boldsymbol{y})}. \tag{1}$$

Recalling that $\Sigma^*$ is a countably infinite set, we cannot expect, in general, to compute Eq. (1) exactly in finite time without additional assumptions.[3] While in some very special cases, e.g., where $p$ and $q$ are deterministic finite-state automata, there exist efficient algorithms, [4, 20, 22], we should not expect such an algorithm to exist in the case where $p$ and $q$ are neural language models, e.g., those based on the transformer [25, 28, 39]. Thus, most researchers turn to approximation, with Monte Carlo estimation being the most widely used method.

The Monte Carlo (MC) estimator for KL divergence (Eq. (1)) involves sampling $M$ strings $\boldsymbol{Y}^{(1)}, \dots, \boldsymbol{Y}^{(M)} \stackrel{\text{i.i.d.}}{\sim} p$ and then averaging $\log \frac{p(\boldsymbol{Y}^{(m)})}{q(\boldsymbol{Y}^{(m)})}$. Even though this estimator is unbiased, it often exhibits high variance, which means the approximation can be noisy and unreliable.[4] More pathologically, the naive MC estimator can result in negative estimates of KL, which may be undesirable in practice.[5] To address these issues, practitioners adopt alternative techniques to ensure non-negativity. For example, Schulman [33] proposes an unbiased, non-negative KL estimator that is widely used in practice [13, 40]. However, the proposed method, in its original form, does not theoretically yield an estimator with lower variance, and, as we show empirically, can exhibit enormous variance.

In this paper, we derive an improved estimator of KL using Rao–Blackwellization [RB; 3, 5, 11, 30], a well-established variance-reduction technique from statistics.[6] This results in an estimator that is provably unbiased *and* has a variance that is always less than or equal to that of the standard Monte Carlo estimator, while requiring no additional computational overhead. As a point of comparison to our RB estimator, we also provide a comprehensive formal analysis of various existing methods for estimating KL divergence, examining their bias and variance.

We empirically validate our theoretical findings using the sentiment-controlled generation task [29] as a testbed. Specifically, we measure the KL divergence between a GPT-2 model [28] before and after fine-tuning, where the fine-tuning objective is to steer the model toward generating positive movie reviews. Our experimental results confirm that our proposed estimator significantly reduces the variance of the Monte Carlo estimator, yielding the most stable and reliable estimates among all methods studied. In contrast, alternative estimators from the literature fail to achieve meaningful variance reduction, and in some cases, lead to unbounded variance. We further examine how using our derived estimator in the fine-tuning loop of RLHF impacts the downstream performance. Our results suggest that using our Rao–Blackwellized estimator reduces the instability across different RLHF runs. We further look

---

[1]Throughout this paper $\log$ denotes the natural logarithm function; thus, KL divergence is measure in *nats* rather than *bits*. We also note that terms of the form $p(\boldsymbol{y}) \log \frac{p(\boldsymbol{y})}{q(\boldsymbol{y})}$ in Eq. (1) where $p(\boldsymbol{y}) = 0$ can *correctly* be taken to equal zero because $\lim_{p \to 0^+} p \log p/q = 0$.

[2]In conditional tasks like dialogue generation, language models are prompted with an input $\boldsymbol{x} \in \Sigma^*$, inducing a conditional distribution $p(\cdot \mid \boldsymbol{x})$. KL divergences are typically averaged over a set of prompts. For simplicity, we omit $\boldsymbol{x}$ in notation and write $p(\boldsymbol{y})$; all of our results extend straightforwardly to the conditional case.

[3]In general, computing the KL divergence between two arbitrary LMs *exactly* is undecidable. To see why, assume that each of the two language models is a probabilistic context-free grammar. In this case, deciding whether their KL divergence is zero is undecidable, as it follows directly from the undecidability of testing equivalence between two unweighted context-free grammars [15]. In the more restrictive case of probabilistic finite-state language models, it is PSPACE-hard. Importantly, however, the intractablity of *exact* computation does not imply that *approximation* is intractable. In practice, one can often obtain good Monte Carlo estimates of the KL divergence, provided that its variance is well-controlled. We study very practical methods for improving variance in this paper.

[4]Monte Carlo estimation formally requires that the underlying random variable have finite variance; if the variance is unbounded, the estimator no longer converges in the limit.

[5]Since the KL divergence is non-negative by definition, a negative estimate can be problematic when KL is used as part of a loss function, as it may destabilize the learning dynamics.

[6]Despite its simplicity, our proposed estimator is absent from existing literature and open-source RLHF libraries [13, 17, 34, 40], highlighting a gap we believe is worth addressing.

at the Pareto frontier of average rewards achieved by the model vs. its KL divergence with the reference model. We observe that models fine-tuned using the RB estimator appear significantly more often on the Pareto frontier of reward vs. KL compared to the models fine-tuned with the MC estimator.

## 2 Preliminaries

### 2.1 Language Models

Let $\Sigma$ be an **alphabet**, a finite, non-empty set of symbols. A **string** is a finite sequence of symbols from $\Sigma$. Let $\Sigma^*$ denote the set of all such strings. A **language model** $p$ is a distribution over $\Sigma^*$. The **prefix probability** function $\vec{p}$ of a prefix $\boldsymbol{x} \in \Sigma^*$ is

$$\vec{p}(\boldsymbol{x}) \overset{\text{def}}{=} \sum_{\boldsymbol{y} \in \Sigma^*} p(\boldsymbol{xy}), \tag{2}$$

which is the cumulative probability of all strings in the language that have $\boldsymbol{x}$ as their prefix. We denote the **conditional prefix probability** as $\vec{p}(\boldsymbol{y} \mid \boldsymbol{x}) = \frac{\vec{p}(\boldsymbol{xy})}{\vec{p}(\boldsymbol{x})}$, where we additionally define $\vec{p}(\text{EOS} \mid \boldsymbol{y}) \overset{\text{def}}{=} \frac{p(\boldsymbol{y})}{\vec{p}(\boldsymbol{y})}$. Language models can be factored into the product of distributions using the chain rule of probability, i.e., for any string $\boldsymbol{y} = y_1 \cdots y_N \in \Sigma^*$ we can write

$$p(\boldsymbol{y}) = \vec{p}(\text{EOS} \mid \boldsymbol{y}) \prod_{n=1}^{N} \vec{p}(y_n \mid \boldsymbol{y}_{<n}), \tag{3}$$

where $\boldsymbol{y}_{<n} \overset{\text{def}}{=} y_1 \cdots y_{n-1}$ and $\text{EOS} \notin \Sigma$ is a distinguished end-of-string symbol. Let $\overline{\Sigma} \overset{\text{def}}{=} \Sigma \cup \{\text{EOS}\}$. In Eq. (3), each $\vec{p}(\cdot \mid \boldsymbol{y}_{<n})$ can fruitfully be viewed as a distribution over $\overline{\Sigma}$. Despite the overloading of the notation, whether $\vec{p}(\cdot \mid \boldsymbol{y}_{<n})$ refers to a prefix probability, a function which takes an argument from $\Sigma^*$, or a distribution over $\overline{\Sigma}$ will always be clear from context. Throughout this paper, we use $\boldsymbol{Y}$ to represent the string-valued random variable sampled from $p$. When taking $M$ i.i.d. samples from $p$, we use $\boldsymbol{Y}^{(m)}$ to denote the $m^{\text{th}}$ sample.

### 2.2 Monte Carlo KL Estimation

A simple way to estimate the KL divergence is with the **Monte Carlo estimator (MC)** defined as

$$\mu_{\text{MC}} = \frac{1}{M} \sum_{m=1}^{M} \log \frac{p(\boldsymbol{Y}^{(m)})}{q(\boldsymbol{Y}^{(m)})} = \frac{1}{M} \sum_{m=1}^{M} f(\boldsymbol{Y}^{(m)}), \tag{4}$$

where $\boldsymbol{Y}^{(1)}, \ldots, \boldsymbol{Y}^{(M)} \overset{\text{i.i.d.}}{\sim} p$ and $f(\boldsymbol{Y}) \overset{\text{def}}{=} \log \frac{p(\boldsymbol{Y})}{q(\boldsymbol{Y})}$. Throughout the paper, we assume that the KL divergence is finite, i.e., $\text{KL}(p \mid\mid q) < \infty$. It is straightforward to show $\mu_{\text{MC}}$ is unbiased, i.e., $\mathbb{E}[\mu_{\text{MC}}] = \text{KL}(p \mid\mid q)$ and the variance of this estimator is $\text{Var}[\mu_{\text{MC}}] = \frac{1}{M} \text{Var}[f(\boldsymbol{Y})]$. In App. A, we discuss the Horvitz–Thompson estimator, another unbiased KL estimator.

Note that while the exact KL value is always non-negative, $f(\boldsymbol{Y})$ may be positive or negative. Consequently, the Monte Carlo estimate $\mu_{\text{MC}}$ may also be negative. This happens because the estimate is based on a limited number of samples, and some sample draws can lead to negative values. This can be problematic during RLHF, which depends on the KL divergence being non-negative.

### 2.3 Control Variate Monte Carlo Estimation

A general approach to reduce estimator variance is through *control variates* [32, §8.2]. For KL divergence between language models, this technique was popularized by Schulman [33] and is widely used in RLHF libraries [13, 17, 34]. Formally, a **control variate** is any function $g \colon \Sigma^* \to \mathbb{R}$ for which $G \overset{\text{def}}{=} \mathbb{E}[g(\boldsymbol{Y})]$ can be efficiently computed.[7] We define the **control variate Monte Carlo estimator** as

$$\mu_{\text{CV}} = \frac{1}{M} \sum_{m=1}^{M} f(\boldsymbol{Y}^{(m)}) + \alpha \cdot (g(\boldsymbol{Y}^{(m)}) - G). \tag{5}$$

---

[7] One could also consider control variates of the form $g \colon \Sigma^* \to \mathbb{R}^d$ for $d > 1$ [10].

where $\alpha \in \mathbb{R}$ is a calibration parameter that must be chosen *a priori*. The proposition below characterizes the variance of $\mu_{\mathrm{CV}}$ as a function of $\alpha$, which will tell use how to choose $\alpha$ optimally.

**Proposition 1.** *Consider the control variate MC estimator $\mu_{\mathrm{CV}}$ defined in Eq. (5), and assume that $\mathbb{E}[g(\boldsymbol{Y})] < \infty$. Then $\mu_{\mathrm{CV}}$ is an unbiased estimator, and its variance is given by*

$$\mathrm{Var}[\mu_{\mathrm{CV}}] = \frac{\mathrm{Var}[f] + \alpha^2 \, \mathrm{Var}[g] + 2\alpha \, \mathrm{Cov}[f, g]}{M}. \tag{6}$$

*Proof.* See App. B. ∎

Assume $0 < \mathrm{Var}[g] < \infty$. It is straightforward to show that $\alpha^* \stackrel{\text{def}}{=} -\,\mathrm{Cov}[f, g]/\mathrm{Var}[g]$ is the value that minimizes the variance. If we plug $\alpha^*$ in Eq. (6), and simplify, we see that

$$\mathrm{Var}[\mu_{\mathrm{CV}}] = \frac{1}{M} \, \mathrm{Var}[f]\Big(1 - \mathrm{Corr}[f, g]^2\Big), \tag{7}$$

which directly translates to reducing the variance of the MC estimator. The magnitude of the correlation between $f$ and $g$ determines the degree of variance reduction. Note that the value of $\alpha^*$ may be estimated from a pilot sample when it cannot be computed analytically.[8]

**KL Estimation with a Control Variate.** A specific control variate for KL estimation was proposed by Schulman [33], who defined $g(\boldsymbol{Y}) = \frac{q(\boldsymbol{Y})}{p(\boldsymbol{Y})}$. Substituting this into Eq. (5), the MC estimator of the KL divergence with this control variate is

$$\mu_{\mathrm{CV}} = \frac{1}{M} \sum_{m=1}^{M} \log \frac{p(\boldsymbol{Y}^{(m)})}{q(\boldsymbol{Y}^{(m)})} + \alpha \cdot \left(\frac{q(\boldsymbol{Y}^{(m)})}{p(\boldsymbol{Y}^{(m)})} - 1\right). \tag{8}$$

**Remarks.** This proposal has some notable properties. First, $G = \mathbb{E}\left[\frac{q(\boldsymbol{Y})}{p(\boldsymbol{Y})}\right] = 1$, meaning $G$ is known in advance.[9] Second, $\mathrm{Cov}[f, g] = 0$ is zero if and only if $p$ is equal to $q$ (Prop. 8), meaning that when the two distributions are not equal, and $\alpha$ is chosen suitably, we are guaranteed to *strictly* reduce variance. Schulman [33] proposes setting $\alpha = 1$; the benefit of this suboptimal choice is that the resulting estimator is always non-negative (Prop. 7).[10] However, setting $\alpha = 1$ will *increase* variance when $\alpha^* < \frac{1}{2}$ (Remark 10). Indeed, our experiments (§5.1) confirm that $\alpha = 1$ is a poor choice in practice—it is better to estimate $\alpha^*$[11] to ensure that the control variate is correctly calibrated.

## 3 Rao–Blackwellized Monte Carlo

In this article, we propose the application of another classical technique to reduce the variance of the Monte Carlo estimation of the KL divergence—**Rao–Blackwellization** [RB; 5, 11]. Despite its standing in the statistics literature, a Rao–Blackwellized Monte Carlo estimator has yet to gain traction in the context of RLHF [13, 17, 34, 40].

We define the **Rao–Blackwellized Monte Carlo estimator** $\mu_{\mathrm{RB}}$ as follows:

$$\mu_{\mathrm{RB}} \stackrel{\text{def}}{=} \frac{1}{M} \sum_{m=1}^{M} \sum_{n=1}^{|\boldsymbol{Y}^{(m)}|} \mathrm{KL}(\vec{p}(\cdot \mid \boldsymbol{Y}_{<n}^{(m)}) \| \vec{q}(\cdot \mid \boldsymbol{Y}_{<n}^{(m)})). \tag{9}$$

The key benefit of this estimator is that we *analytically* compute the expectation over the $n^{\text{th}}$ symbol in each string rather than relying on the single sampled value at that position.

**Rao–Blackwellization Background.** The starting point of Rao–Blackwellization is the following inequality involving the conditional variance: $\mathrm{Var}[\mathbb{E}[\mu \mid T]] \leq \mathrm{Var}[\mu]$, where $\mu$ is an unbiased estimator of $\mathbb{E}[f]$ and $T$ is a statistic[12] for which we can explicitly compute $\mathbb{E}[\mu \mid T]$. This

---

[8]Note that the control variate method can be straightforwardly extended to support multiple control variates.

[9]Note that for this to hold, we need have $q(\boldsymbol{y}) = 0$ whenever $p(\boldsymbol{y}) = 0$, which is different from the support condition we assumed for $\mathrm{KL}(p \,\|\, q) < \infty$.

[10]Note that $\alpha = 1$ is the only value of $\alpha$ that guarantees nonnegativity (see proof of Prop. 7).

[11]Prop. 9 provides the exact conditions on $\alpha$ required for a variance reduction.

[12]Note that $T$ is a function of $\{\boldsymbol{Y}^{(m)}\}_{m=1}^{M}$. We have suppressed this function's arguments in our notation for improved readability.

technique is often referred to as *Rao-–Blackwellization* because the inequality is associated with the Rao–Blackwell theorem [21].[13]

**Notation.** Before moving forward, we introduce some convenient notation. Let $\overline{Y}^{(m)}$ denotes the EOS-padded version of $Y^{(m)}$. Additionally, we extend the definition of the prefix probabilities $\vec{p}$ and $\vec{q}$ to strings containing padding symbols, i.e., $y \notin \Sigma^*$, with the following additional case: $\vec{p}(y \mid \boldsymbol{y}) \stackrel{\text{def}}{=} \mathbb{1}\{y = \text{EOS}\}$, and $\vec{q}(y \mid \boldsymbol{y}) \stackrel{\text{def}}{=} \mathbb{1}\{y = \text{EOS}\}$.

**Understanding Our Rao–Blackwellized Estimator.** We now present a proof that our Rao–Blackwellized estimator $\mu_{\text{RB}}$ is unbiased and indeed does result in a variance reduction. One might wonder why this requires more than a straightforward application of the Rao–Blackwell theorem. The reason is that $\mu_{\text{RB}}$ does not arise directly from the standard formulation of Rao–Blackwellization. Instead, we apply Rao–Blackwellization separately to each summand, where each summand corresponds to an estimator over strings of a particular length. In general, Rao–Blackwellizing the summands pointwise does not guarantee a reduction in the variance of their sum, since the summands may be correlated. However, in the specific case of $\mu_{\text{RB}}$, we *can* prove that the overall variance is reduced, despite Rao–Blackwellizing the summands independently, as we state in the following theorem.

**Theorem 2.** *Suppose the MC estimator $\mu_{\text{MC}}$ has finite variance, i.e., $\text{Var}[\mu_{\text{MC}}] < \infty$. Then the following properties regarding $\mu_{\text{RB}}$ hold:*

$$(i)\ \ \mathbb{E}[\mu_{\text{RB}}] = \text{KL}(p \mid\mid q)\ \ \textit{(unbiasedness)} \qquad (ii)\ \ \text{Var}[\mu_{\text{RB}}] \leq \text{Var}[\mu_{\text{MC}}]\ \ \textit{(variance reduction)}$$

*Proof.* See App. C for a detailed proof. However, we provide the proof sketch below for the reader to quickly understand the structure of the argument, which is broken down into three steps.

(1) **Step-wise Estimation.** We begin by Rao–Blackwellizing the **step-wise Monte Carlo estimator** for any $n > 0$,

$$\mu_{\text{MC}}^n \stackrel{\text{def}}{=} = \frac{1}{M} \sum_{m=1}^{M} \log \frac{\vec{p}(\overline{Y}_n^{(m)} \mid \overline{\boldsymbol{Y}}_{<n}^{(m)})}{\vec{q}(\overline{Y}_n^{(m)} \mid \overline{\boldsymbol{Y}}_{<n}^{(m)})}. \tag{10}$$

Intuitively, $\mu_{\text{MC}}^n$ measures the average KL of the $n^{\text{th}}$ symbol. The next step is to define $T_n(\overline{\boldsymbol{Y}}) = \overline{\boldsymbol{Y}}_{<n}$ and apply Rao–Blackwellization to each $\mu_{\text{MC}}^n$ as follows:

$$\mu_{\text{RB}}^n \stackrel{\text{def}}{=} \mathop{\mathbb{E}}_{\overline{\boldsymbol{Y}}^{(1)},\dots,\overline{\boldsymbol{Y}}^{(M)}} \left[ \mu_{\text{MC}}^n \mid T_n \right] \tag{11a}$$

$$= \frac{1}{M} \sum_{m=1}^{M} \mathop{\mathbb{E}}_{\overline{\boldsymbol{Y}}^{(m)}} \left[ \log \frac{\vec{p}(\overline{Y}_n^{(m)} \mid \overline{\boldsymbol{Y}}_{<n}^{(m)})}{\vec{q}(\overline{Y}_n^{(m)} \mid \overline{\boldsymbol{Y}}_{<n}^{(m)})} \middle| T_n(\overline{\boldsymbol{Y}}^{(m)}) \right] \tag{11b}$$

$$= \frac{1}{M} \sum_{m=1}^{M} \text{KL}(\vec{p}(\cdot \mid \overline{\boldsymbol{Y}}_{<n}^{(m)}) \mid\mid \vec{q}(\cdot \mid \overline{\boldsymbol{Y}}_{<n}^{(m)})). \tag{11c}$$

Now, it is clear from the Rao–Blackwellization theorem that $\mu_{\text{RB}}^n$ is unbiased and it provides a variance reduction (i.e., $\text{Var}[\mu_{\text{RB}}^n] \leq \text{Var}[\mu_{\text{MC}}^n]$) for all $n > 0$. Inuitively, the source of the variance reduction in the $\mu_{\text{RB}}^n$ estimator is that we compute the expectation over the $n^{\text{th}}$ symbol exactly rather relying on the sampled symbol at that position.

(2) **Truncated Estimation.** Next, we define $\mu_{\text{MC}}^{(N)} \stackrel{\text{def}}{=} \sum_{n=1}^{N} \mu_{\text{MC}}^n$ and $\mu_{\text{RB}}^{(N)} \stackrel{\text{def}}{=} \sum_{n=1}^{N} \mu_{\text{RB}}^n$. Intuitively, these estimators target the KL divergence for symbols up to a maximum length of $N$. In Lemma 11, we prove that $\mathbb{E}[\mu_{\text{RB}}^{(N)}] = \mathbb{E}[\mu_{\text{MC}}^{(N)}]$, and $\text{Var}\left[\mu_{\text{RB}}^{(N)}\right] \leq \text{Var}\left[\mu_{\text{MC}}^{(N)}\right]$ for all $N > 0$ using the law of total expectation and Jensen's inequality, where the latter is used in a manner analougsly to have it is used in the original Rao–Blackwellization theorem.

---

[13]Note that in the Rao–Blackwell theorem, we get the stronger result that when $T$ is a sufficient statistic, $\text{Var}[\mathbb{E}[\mu \mid T]]$ is optimal, i.e., it is the minimal-variance, unbiased estimator. However, we can perform Rao–Blackwellization even when $T$ is not sufficient and are still guaranteed that the variance is no worse [31].

(3) **Complete Estimation.** Now, we consider the complete estimation. First, we observe that the limit of the truncated estimators converge to $\mu_{\text{MC}} = \lim_{N \to \infty} \mu_{\text{MC}}^{(N)}$ (Lemma 12), and analogously, $\mu_{\text{RB}} = \lim_{N \to \infty} \mu_{\text{RB}}^{(N)}$. Thereby, we are able to show that $\mu_{\text{RB}}$ is an unbiased estimator of the KL divergence with variance less than or equal to that of $\mu_{\text{MC}}$.

∎

**Remarks.** Notably, $\mu_{\text{RB}}$ is guaranteed to be non-negative, a property desired by some when designing estimators for the KL divergence between two language models, as the KL divergence itself is always non-negative (cf. remarks in §2.3). In the case of our Rao–Blackwellized estimator, nonnonegativity follows from the fact that each step-wise estimator computes the *exact* KL divergence between the two next-symbol distributions, conditioned on the sampled context $\boldsymbol{y}_{<n}^{(m)}$. Since all terms in Eq. (11c) are non-negative, $\mu_{\text{RB}}$ remains non-negative as well.

**Complexity Analysis.** Although computing $\mu_{\text{RB}}$ might seem more expensive than $\mu_{\text{MC}}$, the overall runtime is dominated by the cost of forward passes. Because a forward pass already involves producing the full distribution over $\overline{\Sigma}$ at each position $n$, the additional $\mathcal{O}(MN|\overline{\Sigma}|)$ work required by $\mu_{\text{RB}}$ is negligible compared to the $M$ forward passes needed for both $\mu_{\text{CV}}$ and $\mu_{\text{MC}}$.

## 4 Estimating the Gradient

KL estimation is essential in many applications, especially in fine-tuning large language models. In reinforcement learning from human feedback (RLHF), for example, the objective includes a KL regularization term to balance reward maximization with staying close to a reference model. Since the language model is a differentiable function of parameters $\boldsymbol{\theta}$ optimized via gradient descent, this setup requires computing the gradient of the KL divergence with respect to $\boldsymbol{\theta}$:

$$\boldsymbol{G} \overset{\text{def}}{=} \nabla_{\boldsymbol{\theta}} \text{KL}(p_{\boldsymbol{\theta}} \,||\, q) = \mathbb{E}\left[\log \frac{p_{\boldsymbol{\theta}}(\boldsymbol{Y})}{q(\boldsymbol{Y})} \nabla_{\boldsymbol{\theta}} \log p_{\boldsymbol{\theta}}(\boldsymbol{Y})\right]. \tag{12}$$

Therefore, the Monte Carlo estimator of this gradient is

$$\boldsymbol{\delta}_{\text{MC}} = \frac{1}{M} \sum_{m=1}^{M} \log \frac{p_{\boldsymbol{\theta}}(\boldsymbol{Y}^{(m)})}{q(\boldsymbol{Y}^{(m)})} \nabla_{\boldsymbol{\theta}} \log p_{\boldsymbol{\theta}}(\boldsymbol{Y}^{(m)}). \tag{13}$$

Now, we derive the Rao–Blackwellized Monte Carlo estimator of the gradient. We start with restating Theorem 2.2 in Malagutti et al. [24], which will prove useful.

**Theorem 3** (Malagutti et al. [24]; Theorem 2.2). *Let $p$ and $q$ be language models over $\Sigma$. The following equality holds*

$$\text{KL}(p \,||\, q) = \sum_{\boldsymbol{y} \in \Sigma^*} \vec{p}(\boldsymbol{y}) \text{KL}(\vec{p}(\cdot \mid \boldsymbol{y}) \,||\, \vec{q}(\cdot \mid \boldsymbol{y})), \tag{14}$$

*where we treat $\vec{p}(\cdot \mid \boldsymbol{y})$ and $\vec{q}(\cdot \mid \boldsymbol{y})$ as probability distributions over $\overline{\Sigma}^*$.*

We refer the reader to Malagutti et al. [24] for the proof. Next, to derive the Rao–Blackwellized estimator of the gradient, we take the gradient of the local KL as stated in the following theorem.

**Theorem 4.** *Let $p_{\boldsymbol{\theta}}$ and $q$ be two language models over $\Sigma$ and $\vec{p}_{\boldsymbol{\theta}}$ the prefix probability function of $p_{\boldsymbol{\theta}}$. Then, the following equality holds*

$$\nabla_{\boldsymbol{\theta}} \text{KL}(p_{\boldsymbol{\theta}} \,||\, q) = \sum_{\boldsymbol{y} \in \Sigma^*} \vec{p}_{\boldsymbol{\theta}}(\boldsymbol{y}) \mathbb{E}_Y\left[\log \frac{\vec{p}_{\boldsymbol{\theta}}(Y \mid \boldsymbol{y})}{\vec{q}(Y \mid \boldsymbol{y})} \cdot (\nabla_{\boldsymbol{\theta}} \log \vec{p}_{\boldsymbol{\theta}}(Y \mid \boldsymbol{y}) + \nabla_{\boldsymbol{\theta}} \log \vec{p}_{\boldsymbol{\theta}}(\boldsymbol{y}))\right]. \tag{15}$$

*Proof.* See App. D. ∎

We then construct the Monte Carlo estimator of the gradient using the above theorem, which naturally results in the following unbiased, Rao–Blackwellized Monte Carlo estimator of the gradient:

$$\boldsymbol{\delta}_{\text{RB}} = \frac{1}{M} \sum_{m=1}^{M} \sum_{n=1}^{|\boldsymbol{Y}^{(m)}|} \mathbb{E}_Y\left[\log \frac{\vec{p}_{\boldsymbol{\theta}}(Y \mid \boldsymbol{Y}_{<n}^{(m)})}{\vec{q}(Y \mid \boldsymbol{Y}_{<n}^{(m)})} \cdot \left(\nabla_{\boldsymbol{\theta}} \log \vec{p}_{\boldsymbol{\theta}}(\boldsymbol{Y}_{<n}^{(m)}) + \nabla_{\boldsymbol{\theta}} \log \vec{p}_{\boldsymbol{\theta}}(Y \mid \boldsymbol{Y}_{<n}^{(m)})\right)\right]. \tag{16}$$

The above estimator is unbiased, as it is the MC estimator of Eq. (15).

**Theorem 5.** *Assuming* $\mathrm{Var}[\boldsymbol{\delta}_{\mathrm{RB}}] < \infty, \mathrm{Var}[\boldsymbol{\delta}_{\mathrm{MC}}] < \infty$, *we have*

$$\mathbb{E}\left[\|\boldsymbol{\delta}_{\mathrm{RB}} - \boldsymbol{G}\|^2\right] \leq \mathbb{E}\left[\|\boldsymbol{\delta}_{\mathrm{MC}} - \boldsymbol{G}\|^2\right]. \tag{17}$$

*Proof.* See App. D.1. ∎

**Off-policy Gradient.** So far, we have discussed how to estimate the gradient of $\mathrm{KL}(p_{\boldsymbol{\theta}} \| q)$ using samples drawn from the current policy $p_{\boldsymbol{\theta}}$. Crucially, we derive the gradient manually rather than relying on automatic differentiation because the samples depend on $\boldsymbol{\theta}$ through $p_{\boldsymbol{\theta}}$. However, in practice and for efficiency reasons, we often collect large batches of samples in parallel with the optimization loop. As a result, these samples are generated from a slightly outdated version of the policy, denoted $p_{\boldsymbol{\theta}_{\mathrm{old}}}$. To compute the KL divergence using samples from $p_{\boldsymbol{\theta}_{\mathrm{old}}}$, we first write the KL as the expectation under $p_{\boldsymbol{\theta}_{\mathrm{old}}}$ as

$$\mathrm{KL}(p_{\boldsymbol{\theta}} \| q) = \underset{\boldsymbol{Y} \sim p_{\boldsymbol{\theta}_{\mathrm{old}}}}{\mathbb{E}}\left[\frac{p_{\boldsymbol{\theta}}(\boldsymbol{Y})}{p_{\boldsymbol{\theta}_{\mathrm{old}}}(\boldsymbol{Y})} \log \frac{p_{\boldsymbol{\theta}}(\boldsymbol{Y})}{q(\boldsymbol{Y})}\right]. \tag{18}$$

Therefore, the MC estimator using samples $\boldsymbol{Y}^{(1)}, \dots, \boldsymbol{Y}^{(M)} \overset{\text{i.i.d.}}{\sim} p_{\boldsymbol{\theta}_{\mathrm{old}}}$ is

$$\mu_{\mathrm{MC}}^{\mathrm{old}} = \frac{1}{M}\sum_{m=1}^{M}\frac{p_{\boldsymbol{\theta}}(\boldsymbol{Y}^{(m)})}{p_{\boldsymbol{\theta}_{\mathrm{old}}}(\boldsymbol{Y}^{(m)})} \log \frac{p_{\boldsymbol{\theta}}(\boldsymbol{Y}^{(m)})}{q(\boldsymbol{Y}^{(m)})}. \tag{19}$$

Given the unbiasedness proof of the Rao–Blackwellization in Thm. 2, we can similarly write

$$\mathrm{KL}(p_{\boldsymbol{\theta}} \| q) = \underset{\boldsymbol{Y} \sim p_{\boldsymbol{\theta}_{\mathrm{old}}}}{\mathbb{E}}\left[\frac{p_{\boldsymbol{\theta}}(\boldsymbol{Y})}{p_{\boldsymbol{\theta}_{\mathrm{old}}}(\boldsymbol{Y})} \sum_{n=1}^{|\boldsymbol{Y}^{(m)}|} \underset{Y}{\mathbb{E}}\left[\log \frac{\vec{p}_{\boldsymbol{\theta}}(Y \mid \boldsymbol{Y}_{<n})}{\vec{q}(Y \mid \boldsymbol{Y}_{<n})}\right]\right]. \tag{20}$$

Therefore, the Rao–Blackwellized MC estimator using samples $\boldsymbol{Y}^{(1)}, \dots, \boldsymbol{Y}^{(M)} \overset{\text{i.i.d.}}{\sim} p_{\boldsymbol{\theta}_{\mathrm{old}}}$ is

$$\mu_{\mathrm{RB}}^{\mathrm{old}} = \frac{1}{M}\sum_{m=1}^{M}\frac{p_{\boldsymbol{\theta}}(\boldsymbol{Y}^{(m)})}{p_{\boldsymbol{\theta}_{\mathrm{old}}}(\boldsymbol{Y}^{(m)})} \sum_{n=1}^{|\boldsymbol{Y}^{(m)}|} \underset{Y}{\mathbb{E}}\left[\log \frac{\vec{p}_{\boldsymbol{\theta}}(Y \mid \boldsymbol{Y}_{<n}^{(m)})}{\vec{q}(Y \mid \boldsymbol{Y}_{<n}^{(m)})}\right]. \tag{21}$$

Since $\mu_{\mathrm{MC}}^{\mathrm{old}}$ and $\mu_{\mathrm{RB}}^{\mathrm{old}}$ use samples from the old policy that does not depend on $\boldsymbol{\theta}$, we can apply automatic differentiation to compute the estimate of the KL gradient by computing the gradient of $\mu_{\mathrm{MC}}^{\mathrm{old}}$ and $\mu_{\mathrm{RB}}^{\mathrm{old}}$.

## 5 Experiments

We use the sentiment control task as the testbed to empirically evaluate our theoretical findings on the KL estimators. Concretely, the reference model, denoted as $q$, is the `GPT-IMDB`[14] model, i.e., a `GPT-2` [28] model fine-tuned on IMDB corpus [23]. The goal of the task is to fine-tune this language model such that the samples from it are movie reviews with a positive sentiment. The fine-tuned language model is denoted with $p_{\boldsymbol{\theta}}$. In the following experiments, we estimate the KL divergence between $p_{\boldsymbol{\theta}}$ and $q$. We provide a code snippet for implementing the RB estimator in App. F.1.

### 5.1 Analyzing the KL Estimators

In this experiment, we empirically evaluate the bias, variance, and consistency of various KL estimators. To obtain $p_{\boldsymbol{\theta}}$, we fine-tune $q$ with direct preference optimization [DPO; 29] on a sample of 5,000 data points from the IMDB training set. To create the preference data required for DPO training, following Rafailov et al. [29], we sample 4 responses for each prompt and create 6 pairs per prompt.

---

[14]Specifically, we use `https://huggingface.co/lvwerra/gpt2-imdb`.

To determine the preferred response in each pair, we employ a binary sentiment classifier,[15] selecting the response with the higher probability of positive sentiment. Upon successful fine-tuning, $p_\theta$ should assign a higher probability mass to movie reviews with positive sentiment while maintaining a low KL divergence with $q$. We then evaluate this KL divergence using our estimators to assess their reliability in measuring distributional shifts induced by fine-tuning.

We evaluate on 512 examples from the IMDB dataset. For each review, we randomly select a prefix length between 2 and 8 tokens and use it as the prompt. we then generate 4000 samples from $p_\theta$ for each prompt. Using these samples, we compute the MC, control variate (CV), and Rao–Blackwellized (RB) estimators and estimate their standard We also implement the Horvitz–Thompson (HT) estimator; see App. A for details. The CV estimator, $\mu_{\mathrm{CV}}$, is computed twice: once using the optimal $\alpha$ estimated from 1000 samples, and once with $\alpha = 1$ to match the setup in Schulman [33].

Table 1: Estimated value $\pm$ empirical standard deviation of different estimators. When aggregating over prompts, $\mu_{\mathrm{HT}}$ and $\mu_{\mathrm{CV}}$ fail to significantly reduce the variance of $\mu_{\mathrm{MC}}$. RB estimator, however, achieves the lowest standard deviation.

|  | $M = 1$ | $M = 5$ | $M = 10$ |
|---|---|---|---|
| $\mu_{\mathrm{MC}}$ | $6.76 \pm 0.16$ | $6.76 \pm 0.07$ | $6.76 \pm 0.05$ |
| $\mu_{\mathrm{HT}}$ | $6.76 \pm 0.16$ | $6.76 \pm 0.07$ | $6.76 \pm 0.05$ |
| $\mu_{\mathrm{CV1}}$ | $6.28 \pm 2.54$ | $6.28 \pm 1.13$ | $6.28 \pm 0.79$ |
| $\mu_{\mathrm{CV}}$ | $6.76 \pm 0.16$ | $6.76 \pm 0.07$ | $6.76 \pm 0.05$ |
| $\mu_{\mathrm{RB}}$ | $6.76 \pm \mathbf{0.11}$ | $6.76 \pm \mathbf{0.05}$ | $6.76 \pm \mathbf{0.03}$ |

In Tab. 1, we report the expected KL estimate along with the empirical standard deviation of different estimators evaluated at sample sizes $M = 1, 5, 10$. To obtain these estimates, we compute each estimator using $M$ samples, repeating the process $4000/M$ times to estimate both the expected value and the standard deviation of the estimates. Our findings confirm that all estimators except one ($\mu_{\mathrm{CV}}, \alpha = 1$), are unbiased and report an expected KL divergence of 6.76. We also observe that the CV estimators fail to significantly reduce the variance of the standard MC estimator. Importantly, the RB estimator achieves the lowest standard deviation and offers a more robust estimate compared to

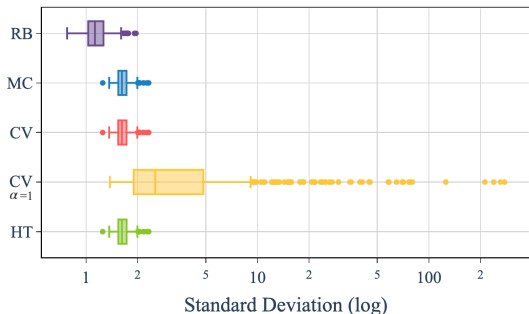

Figure 1: Standard deviation of KL estimators across various prompts in the IMDB datasest.

the MC estimator. Interestingly, we observe that the $\mu_{\mathrm{CV}}$ estimator exhibits a noticeable bias and high standard deviation when $\alpha = 1$, i.e., when it is not set to its optimal value. The bias arises from numerical instability during the computation of $g(\boldsymbol{Y}) = \frac{q(\boldsymbol{Y})}{p(\boldsymbol{Y})} - 1$. The high variance is due to large values of $\mathrm{Var}[g(\boldsymbol{Y})]$. Specifically, for certain prompts, $\mathrm{Var}[g(\boldsymbol{Y})]$ can be unbounded. We visualize the estimates for 3 example prompts in App. E.

Since the robustness of the estimators depends on the choice of the prompt, we further analyze their estimated standard deviations across all prompts. Fig. 1 presents a box plot of standard deviations (in log scale) for each estimator. The $\mu_{\mathrm{CV}}$ estimator with $\alpha = 1$ shows significant instability for certain prompts, with numerous outliers indicating high variance. In contrast, the $\mu_{\mathrm{MC}}$ and the standard $\mu_{\mathrm{CV}}$ estimators exhibit comparable standard deviations. In particular, the Rao-Blackwellized estimator consistently achieves the lowest standard deviation, suggesting that it provides the most stable estimates.

## 5.2 KL Estimation and RLHF Training Dynamics

A key application of KL estimation is in the RLHF training loop. From the previous experiment, we observed that the RB estimator significantly reduces the standard deviation of the MC estimator. Therefore, it is natural to ask how this affects RLHF performance when this estimator is used in the training loop. The RLHF objective consists of two terms: (i) the expected rewards for samples generated by the language model $p_\theta$, which in this case is the samples' score under a sentiment classifier[16], and (ii) the negative KL divergence between the fine-tuned model $p_\theta$ and the reference model $q$, which represents the language model before fine-tuning.

---

[15]Specifically, we use https://huggingface.co/lvwerra/distilbert-imdb.
[16]Specifically, we look at the logits of the positive class.

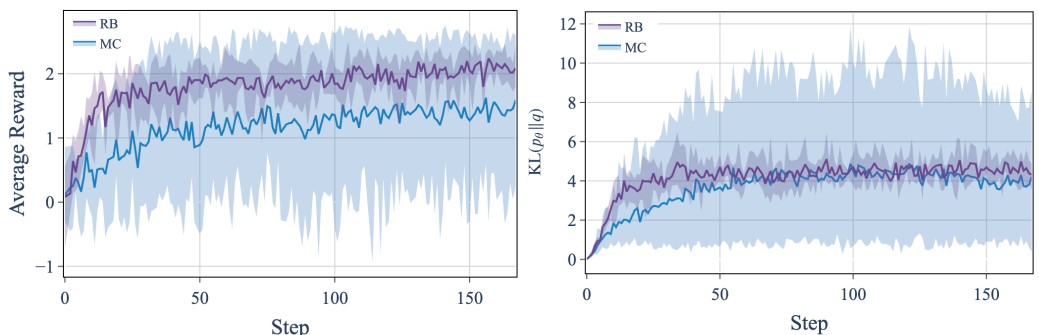

Figure 2: Comparison of the Monte Carlo (MC) and Rao–Blackwellized (RB) estimators in the RLHF fine-tuning loop. We perform RLHF with each estimator 5 times and plot the mean and standard deviation (in shades) of the average reward values and the KL at each fine-tuning step. We observe that the MC estimator is not as stable as the RB estimator and its performance varies significantly across different runs. However, RB estimator reliably offers a good balance between achieving low KL and high reward values in all runs.

We compare the MC and RB estimators for computing the gradient of the KL divergence term in the RLHF training loop. We use the RL algorithm[17] proposed by Ahmadian et al. [2],[18] which is an improved verfaiion of the REINFORCE algorithm [41].[19]

First, we empirically test Thm. 5 by measuring the variance of the gradient norm. We sample 40 prompts and compute the gradient of the KL divergence with respect to the model parameters using both the MC and RB estimators. To estimate variance empirically, we repeat this process 5 times. Both estimators are evaluated on the same prompts and model initializations to ensure a fair comparison. We find that the variance of the gradient norm estimated with the MC estimator is 59.90, whereas with the RB estimator it is 45.44. This corresponds to a 24.6% reduction in variance, providing direct empirical evidence consistent with our theoretical motivation.

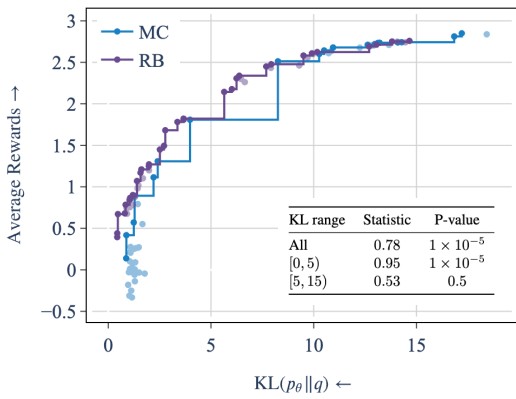

Figure 3: Compared to models trained with MC esimator, models trained with RB appear on the Pareto front 78% of the time.

We then proceed with using both estimators in RL fine-tuning. We track two metrics throughout fine-tuning: (i) the average reward associated with samples drawn from $p_{\boldsymbol{\theta}}$, and (ii) the KL divergence between $p_{\boldsymbol{\theta}}$ and $q$. The results are visualized in Fig. 2. The purple trace represents the training run where the $\mu_{\mathrm{RB}}$ is used in the optimization loop to estimate the gradient of the KL divergence, while the blue trace represents the run using $\mu_{\mathrm{MC}}$. The x-axis denotes the fine-tuning step, with the left plot showing the evolution of the average reward and the right plot displaying the KL divergence between $p_{\boldsymbol{\theta}}$ and $q$ over the course of fine-tuning. We repeat each experiment 5 times and report the mean and standard deviation of each metric. Notably, the KL values in the right plot are estimated using the RB estimator. However, we observe the same overall trend when using the MC estimator for evaluation.

As illustrated in Fig. 2, the performance of the models trained using the MC estimator varies significantly across the 5 experiments, resulting in a large standard deviation in both average rewards and the KL divergence. However, RB estimator consistently achieves high rewards and reasonable KL values across all runs. This observation suggests that the RB estimator makes RLHF runs more stable.

---

[17]App. D.2 discusses a common mistake when Rao–Blackwellizing the KL estimator in trust-region algorithms.
[18]Specifically, we use the available implementation of this algorithm in the `trl` library [40].
[19]App. F reports the hyperparameters used for the algorithm.

Finally, we vary the KL coefficient, $\beta$, in $[0.01, 0.1]$ range and fine-tune 18 models with each estimator. For each estimator, we plot the Pareto frontier of average rewards versus KL divergence in Fig. 3, displaying models that do not appear on the Pareto front with reduced opacity. Overall, we find that fine-tuning with the RB estimator is more effective at achieving high rewards while maintaining a low KL divergence from the reference model. To quantify this effect, we compute the fraction of RB fine-tuned models that appear on the overall Pareto front—i.e., the frontier obtained when considering all models fine-tuned with either estimator. We then conduct a permutation test and report the results in Fig. 3. We find that $78\%$ of the points on the overall Pareto front come from RB fine-tuned models. Restricting to models with KL values below 5, this fraction rises to $95\%$, with both results being statistically significant.

## 6 Conclusion

In this paper, we study the problem of estimating the KL divergence between language models. We provide a comprehensive formal analysis of various KL estimators, with a focus on their bias and variance. We introduce the RB estimator, which is provably unbiased and has variance at most equal to that of the standard NC estimator. This estimator applies the well-known Rao–Blackwellization technique to reduce the variance of the standard MC method. Our empirical results show that the RB estimator significantly reduces the variance compared to the MC estimator, while other estimators fail to achieve meaningful variance reduction or, in some cases, suffer from unbounded variance. Additionally, we find that using our proposed RB estimator makes RLHF more stable and produces models that more frequently lie on the Pareto frontier of reward versus KL, compared to models fine-tuned with the MC estimator.

## Impact Statement

In this paper, we investigate the fundamental problem of estimating KL divergence between language models. One key application of KL estimation is in RLHF, which aims to enhance fluency while aligning language models with user preferences. However, RLHF can also be misused by bad actors to optimize models for generating misleading, biased, or harmful content. While our work provides a deeper understanding of KL estimation techniques, it is purely foundational research and does not introduce new risks or directly contribute to harmful applications.

## Limitations

In our RLHF experiments, evaluating the variance of our estimator and comparing it to existing methods requires training a large number of models. For instance, the significance test in §5.2 involves training 36 models. Due to limited computational resources, we used the controlled-generation task as a testbed. Given the strength of both our theoretical and empirical results, we hope future work will adopt the Rao–Blackwellized estimator and apply it to larger language models and a wider variety of RL-based approaches to LM alignment.

## Acknowledgements

We thank Ahmad Beirami and Cristina Pinneri for the insightful discussions throughout the course of this project. We also thank Alexander K. Lew for the valuable feedback on a draft of this paper. Afra Amini is supported by the ETH AI Center doctoral fellowship.

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

## A Horvitz–Thompson Estimation

When estimating the KL divergence between language models $p$ and $q$, we have access not only to samples from $p$ but also to the probability assigned by $p$ to any string $\boldsymbol{y}$. This enables the use of a more informed estimator, which leverages these probabilities during its construction. Notably, this estimator is a specific instance of the **Horvitz–Thompson (HT)** estimator [16], defined as

$$\mu_{\text{HT}} = \sum_{\boldsymbol{y} \in S} \frac{p(\boldsymbol{y})}{\pi_S(\boldsymbol{y})} \log \frac{p(\boldsymbol{y})}{q(\boldsymbol{y})} = \sum_{\boldsymbol{y} \in S} \frac{p(\boldsymbol{y})}{\pi_S(\boldsymbol{y})} f(\boldsymbol{y}), \tag{22}$$

where $S$ is the random variable representing the *set* of all sampled strings. Any sampling design can be specified to generate the elements of $S$. The **inclusion probability**, denoted by $\pi_S(\boldsymbol{y})$, is the probability that a particular string $\boldsymbol{y}$ is included in $S$, or equivalently, $\mathbb{E}\left[\mathbb{1}\{\boldsymbol{y} \in S\}\right]$.

**Proposition 6.** $\mu_{\text{HT}}$ *is an unbiased estimator of the KL divergence, i.e.,* $\mathbb{E}\left[\mu_{\text{HT}}\right] = \text{KL}(p \mid\mid q)$.

*Proof.* The bias of the estimator is as follows:

$$\mathbb{E}_S \left[\mu_{\text{HT}}\right] - \text{KL}(p \mid\mid q)$$

$$= \mathbb{E}_S \left[ \sum_{\boldsymbol{y} \in S} \frac{p(\boldsymbol{y})}{\pi_S(\boldsymbol{y})} \log \frac{p(\boldsymbol{y})}{q(\boldsymbol{y})} \right] - \text{KL}(p \mid\mid q) \qquad \text{(definition of } \mu_{\text{HT}}\text{)} \tag{23a}$$

$$= \mathbb{E}_S \left[ \sum_{\boldsymbol{y} \in \Sigma^*} \frac{p(\boldsymbol{y})}{\pi_S(\boldsymbol{y})} \log \frac{p(\boldsymbol{y})}{q(\boldsymbol{y})} \cdot \mathbb{1}\{\boldsymbol{y} \in S\} \right] - \text{KL}(p \mid\mid q) \tag{23b}$$

$$= \sum_{\boldsymbol{y} \in \Sigma^*} \frac{p(\boldsymbol{y})}{\pi_S(\boldsymbol{y})} \log \frac{p(\boldsymbol{y})}{q(\boldsymbol{y})} \cdot \mathbb{E}_S \left[ \mathbb{1}\{\boldsymbol{y} \in S\} \right] - \text{KL}(p \mid\mid q) \qquad \text{(linearity of expectation)} \tag{23c}$$

$$= \sum_{\boldsymbol{y} \in \Sigma^*} \frac{p(\boldsymbol{y})}{\cancel{\pi_S(\boldsymbol{y})}} \log \frac{p(\boldsymbol{y})}{q(\boldsymbol{y})} \cdot \cancel{\pi_S(\boldsymbol{y})} - \text{KL}(p \mid\mid q) \qquad \text{(definition of } \pi_S(\boldsymbol{y})\text{)} \tag{23d}$$

$$= 0. \tag{23e}$$

∎

Similar to the MC estimator, the HT estimator does not necessarily return a non-negative estimate of the KL. In principle, however, we should prefer Eq. (22) to Eq. (4) because it exploits more information—namely, the knowledge of $p$. Whether the HT estimator yields lower variance than the MC estimator depends on the sampling design used to construct $S$. In our experiments in App. E, we used the sampling-with-replacement design, where $\pi_S(\boldsymbol{y}) = 1 - (1 - p(\boldsymbol{y}))^M$. Compared to the MC estimator, we observed no significant reduction in variance in our experiments.

# B    Control Variate Monte Carlo Estimation

**Proposition 1.** *Consider the control variate MC estimator $\mu_{\mathrm{CV}}$ defined in Eq. (5), and assume that $\mathbb{E}[g(\boldsymbol{Y})] < \infty$. Then $\mu_{\mathrm{CV}}$ is an unbiased estimator, and its variance is given by*

$$\mathrm{Var}[\mu_{\mathrm{CV}}] = \frac{\mathrm{Var}[f] + \alpha^2 \, \mathrm{Var}[g] + 2\alpha \, \mathrm{Cov}[f, g]}{M}. \tag{6}$$

*Proof.* Recall that the control variate Monte Carlo estimator is defined as

$$\mu_{\mathrm{CV}} = \frac{1}{M} \sum_{m=1}^{M} f(\boldsymbol{Y}^{(m)}) + \alpha \cdot (g(\boldsymbol{Y}^{(m)}) - G). \tag{5}$$

Note that $\{\boldsymbol{Y}^{(m)}\}_{m=1}^{M} \overset{\text{i.i.d.}}{\sim} p$. First, we look at the expected value of the estimator:

$$\mathbb{E}[\mu_{\mathrm{CV}}] = \frac{1}{M} \sum_{m=1}^{M} \mathbb{E}\Big[f(\boldsymbol{Y}^{(m)})\Big] + \alpha \cdot \underbrace{(\mathbb{E}\Big[g(\boldsymbol{Y}^{(m)})\Big] - G)}_{=0} = \mathbb{E}[f] \tag{24a}$$

Therefore, it is unbiased. Next, we manipulate the variance

$$\mathrm{Var}[\mu_{\mathrm{CV}}] = \frac{1}{M} \mathrm{Var}[f + \alpha \cdot (g - G)] = \frac{1}{M} \mathrm{Var}[f] + \frac{\alpha^2}{M} \mathrm{Var}[g] + \frac{2\alpha}{M} \mathrm{Cov}[f, g], \tag{24b}$$

where the second equality above stems from well-known variance identities.[20]    ∎

**Proposition 7.** *Provided that the target KL divergence is finite, the estimator $\mu_{\mathrm{CV}}$ with $\alpha = 1$ is nonnegative with probability one. Furthermore, $\alpha = 1$ is the only value for which we can guarantee nonnegativity of $\mu_{\mathrm{CV}}$.*

*Proof.* To prove that $\mu_{\mathrm{CV}}$ is nonnegative with probability one when $\alpha = 1$, we will prove that each term $\log \frac{p(\boldsymbol{Y}^{(m)})}{q(\boldsymbol{Y}^{(m)})} + \alpha \cdot \left(\frac{q(\boldsymbol{Y}^{(m)})}{p(\boldsymbol{Y}^{(m)})} - 1\right)$ in the summation that defines $\mu_{\mathrm{CV}}$ is nonnegative.

Let $r = \frac{q(\boldsymbol{Y}^{(m)})}{p(\boldsymbol{Y}^{(m)})}$. We identify $\alpha$ for which $\log \frac{1}{r} + \alpha \cdot (r - 1) \geq 0$ holds for all $r > 0$.[21] Equivalently, we want to find $\alpha$ values for which:

$$\inf_{r > 0} - \log r + \alpha \cdot (r - 1) \geq 0. \tag{25a}$$

Next, we prove that $\alpha = 1$ is the only value satisfies the inequality Eq. (25a). To see why, consider the first-order optimality conditions for the minimization over $r$. These conditions are necessary and sufficient as $-\log r + \alpha \cdot (r - 1)$ is convex for all $\alpha$ over the region $r > 0$. Solving for $r$ such that the derivative is zero gives us

$$0 = \frac{\partial}{\partial r} \left[ -\log r + \alpha \cdot (r - 1) \right] \tag{25b}$$

$$\iff 0 = -1/r + \alpha \tag{25c}$$

$$\iff r = 1/\alpha. \tag{25d}$$

Plugging that value allows us to simplify away the minimization:

$$0 \leq \inf_{r > 0} - \log r + \alpha \cdot (r - 1) \tag{25e}$$

$$= -\log(1/\alpha) + \alpha \cdot (1/\alpha - 1) \tag{25f}$$

$$= \log(\alpha) + 1 - \alpha. \tag{25g}$$

For each term in $\mu_{\mathrm{CV}}$ to be nonnegative, we need $\log(\alpha) + 1 - \alpha \geq 0$. However, the function $\log(\alpha) + 1 - \alpha$ is strictly concave and nonpositive for all $\alpha > 0$, attaining zero only at $\alpha = 1$. Thus, when $\alpha = 1$ every term in $\mu_{\mathrm{CV}}$ is nonnegative. When $\alpha \neq 1$, some terms may be nonnegative, leading to the possibility that $\mu_{\mathrm{CV}}$ may be negative.

∎

---

[20]Let $X$ and $Y$ be real-valued random variables, and let $a$ be a real scalar. Then, the following hold: $\mathrm{Var}[X + Y] = \mathrm{Var}[X] + \mathrm{Var}[Y] + 2 \, \mathrm{Cov}[X, Y]$, $\mathrm{Var}[aX] = a^2 \, \mathrm{Var}[X]$, and $\mathrm{Var}[X + a] = \mathrm{Var}[X]$.
[21]Note that $r > 0$ whenever this term is finite.

**Proposition 8.** *Given the random variable $\boldsymbol{Y} \sim p$, let $f(\boldsymbol{Y}) = \log \frac{p(\boldsymbol{Y})}{q(\boldsymbol{Y})}$ and $g(\boldsymbol{Y}) = \frac{q(\boldsymbol{Y})}{p(\boldsymbol{Y})}$. We prove $\mathrm{Cov}[f, g]$ is zero if and only if $p = q$.*

*Proof.*

$$\mathrm{Cov}[f,g] = \mathbb{E}\left[\left(\log \frac{p(\boldsymbol{Y})}{q(\boldsymbol{Y})} - \mathbb{E}\left[\log \frac{p(\boldsymbol{Y})}{q(\boldsymbol{Y})}\right]\right)\left(\frac{q(\boldsymbol{Y})}{p(\boldsymbol{Y})} - \underbrace{\mathbb{E}\left[\frac{q(\boldsymbol{Y})}{p(\boldsymbol{Y})}\right]}_{=1}\right)\right] \tag{26a}$$

$$= \mathbb{E}\left[\frac{q(\boldsymbol{Y})}{p(\boldsymbol{Y})} \log \frac{p(\boldsymbol{Y})}{q(\boldsymbol{Y})} - \log \frac{p(\boldsymbol{Y})}{q(\boldsymbol{Y})} - \mathbb{E}\left[\log \frac{p(\boldsymbol{Y})}{q(\boldsymbol{Y})}\right]\frac{q(\boldsymbol{Y})}{p(\boldsymbol{Y})} + \mathbb{E}\left[\log \frac{p(\boldsymbol{Y})}{q(\boldsymbol{Y})}\right]\right] \tag{26b}$$

$$= \mathbb{E}\left[\frac{q(\boldsymbol{Y})}{p(\boldsymbol{Y})} \log \frac{p(\boldsymbol{Y})}{q(\boldsymbol{Y})}\right] - \mathbb{E}\left[\log \frac{p(\boldsymbol{Y})}{q(\boldsymbol{Y})}\right]\underbrace{\mathbb{E}\left[\log \frac{q(\boldsymbol{Y})}{p(\boldsymbol{Y})}\right]}_{=1} \tag{26c}$$

$$= \mathbb{E}\left[-\frac{q(\boldsymbol{Y})}{p(\boldsymbol{Y})} \log \frac{q(\boldsymbol{Y})}{p(\boldsymbol{Y})}\right] - \mathbb{E}\left[\log \frac{p(\boldsymbol{Y})}{q(\boldsymbol{Y})}\right] \tag{26d}$$

$$= -\mathrm{KL}(q \parallel p) - \mathrm{KL}(p \parallel q). \tag{26e}$$

Since each KL divergence is non-negative and equals zero if and only if the two distributions coincide almost everywhere, we have,

$$\mathrm{Cov}[f,g] = 0 \iff -\mathrm{KL}(q \parallel p) - \mathrm{KL}(p \parallel q) = 0 \iff p = q. \tag{26f}$$

$\blacksquare$

**Proposition 9.**

$$\mathrm{Var}\left[\mu_{\mathrm{CV}}^{(\alpha)}\right] \le \mathrm{Var}\left[\mu_{\mathrm{MC}}\right] \iff \alpha \in [\min(0, 2\alpha^*), \max(0, 2\alpha^*)] \tag{27}$$

*Proof.* We first establish conditions of $\alpha$ for which the variance of $\mathrm{Var}[\mu_{\mathrm{CV}}]$ does not increase that of $\mathrm{Var}[\mu_{\mathrm{MC}}]$. In other words, we seek conditions on $\alpha$ such that the following inequality holds:

$$\mathrm{Var}\left[\mu_{\mathrm{CV}}^{(\alpha)}\right] \le \mathrm{Var}\left[\mu_{\mathrm{MC}}\right] \tag{28}$$

$$\frac{\mathrm{Var}[f] + \alpha^2 \mathrm{Var}[g] + 2\alpha \mathrm{Cov}[f,g]}{M} \le \frac{\mathrm{Var}[f]}{M} \tag{29}$$

$$\mathrm{Var}[f] + \alpha^2 \mathrm{Var}[g] + 2\alpha \mathrm{Cov}[f,g] \le \mathrm{Var}[f] \tag{30}$$

$$\alpha^2 \mathrm{Var}[g] + 2\alpha \mathrm{Cov}[f,g] \le 0. \tag{31}$$

Observe that the function on the left-hand side is quadratic in $\alpha$, and, moreover, it is *convex* because $\mathrm{Var}[g] \ge 0$. The minimum of this quadratic is $\alpha^* = -\frac{\mathrm{Cov}[f,g]}{\mathrm{Var}[g]}$. We can use the quadratic formula to identify the two values of $\alpha$ where it equals 0, i.e., $\alpha \in \{0, 2\alpha^*\}$. Now, because the quadratic is convex, we have that it is $\le 0$ for values of $\alpha \in [\min(0, 2\alpha^*), \max(0, 2\alpha^*)]$. Note that $\alpha^*$ can be positive or negative; hence the min and max. $\blacksquare$

**Remark 10.** *When does Schulman's (2020) suboptimal choice of $\alpha = 1$ not hurt estimator variance? To answer this question, we substitute $\alpha = 1$ into the variance of $\mu_{\mathrm{CV}}$ given in Eq. (6), we have*

$$\mathrm{Var}[\mu_{\mathrm{CV}}] = \frac{\mathrm{Var}[f] + \mathrm{Var}[g] + 2\mathrm{Cov}[f,g]}{M}. \tag{32}$$

*Therefore, for $\mathrm{Var}[\mu_{\mathrm{CV}}] \le \mathrm{Var}[\mu_{\mathrm{MC}}]$, we must have*

$$\frac{\mathrm{Var}[f] + \mathrm{Var}[g] + 2\mathrm{Cov}[f,g]}{M} \le \frac{\mathrm{Var}[f]}{M} \tag{33a}$$

$$\mathrm{Var}[f] + \mathrm{Var}[g] + 2\mathrm{Cov}[f,g] \le \mathrm{Var}[f] \tag{33b}$$

$$\mathrm{Var}[g] + 2\,\mathrm{Cov}[f, g] \leq 0 \tag{33c}$$

$$-\frac{\mathrm{Cov}[f, g]}{\mathrm{Var}[g]} \geq \frac{1}{2} \tag{33d}$$

$$\alpha^* \geq \frac{1}{2}. \tag{33e}$$

*Therefore, choosing $\alpha = 1$ does not hurt the variance if $\alpha^* \geq \frac{1}{2}$, but does otherwise.*

## C    Rao–Blackwellized Estimator

**Lemma 11.** *With regard to the estimator $\mu_{\mathrm{RB}}^{(N)}$, the following two properties hold for all $N > 0$*

*1.* $\mathbb{E}[\mu_{\mathrm{RB}}^{(N)}] = \mathbb{E}[\mu_{\mathrm{MC}}^{(N)}]$                                      *(unbiasedness)*

*2.* $\mathrm{Var}\left[\mu_{\mathrm{RB}}^{(N)}\right] \leq \mathrm{Var}\left[\mu_{\mathrm{MC}}^{(N)}\right]$                      *(variance reduction)*

*Proof.* In the proof below, we consider the case where $M = 1$. The proof easily generalizes to $M > 1$ using the i.i.d. assumption. Additionally, we write $\mu_{\mathrm{RB}}^{(N)}(\overline{\boldsymbol{Y}})$, rather than suppressing the argument to the estimator as we do in the main text. We begin by proving the unbiasedness property of the estimator.

$$\mathbb{E}[\mu_{\mathrm{RB}}^{(N)}] = \mathop{\mathbb{E}}_{\overline{\boldsymbol{Y}}'} \left[ \sum_{n=1}^{N} \mathop{\mathbb{E}}_{\overline{\boldsymbol{Y}}} \left[ \mu_{\mathrm{MC}}^{n}(\overline{\boldsymbol{Y}}) \mid \overline{\boldsymbol{Y}}_{<n} = \overline{\boldsymbol{Y}}'_{<n} \right] \right] \qquad \text{(definition of } \mu_{\mathrm{RB}}^{(N)}) \qquad (34a)$$

$$= \sum_{n=1}^{N} \mathop{\mathbb{E}}_{\overline{\boldsymbol{Y}}'} \left[ \mathop{\mathbb{E}}_{\overline{\boldsymbol{Y}}} \left[ \mu_{\mathrm{MC}}^{n}(\overline{\boldsymbol{Y}}) \mid \overline{\boldsymbol{Y}}_{<n} = \overline{\boldsymbol{Y}}'_{<n} \right] \right] \qquad \text{(linearity of expectation)} \qquad (34b)$$

$$= \sum_{n=1}^{N} \mathop{\mathbb{E}}_{\overline{\boldsymbol{Y}}} \left[ \mu_{\mathrm{MC}}^{n}(\overline{\boldsymbol{Y}}) \right] \qquad \text{(law of total expectation)} \qquad (34c)$$

$$= \mathop{\mathbb{E}}_{\overline{\boldsymbol{Y}}} \left[ \sum_{n=1}^{N} \mu_{\mathrm{MC}}^{n}(\overline{\boldsymbol{Y}}) \right] \qquad \text{(linearity of expectation)} \qquad (34d)$$

$$= \mathbb{E}[\mu_{\mathrm{MC}}^{(N)}] \qquad \text{(definition of } \mu_{\mathrm{MC}}^{(N)}) \qquad (34e)$$

Next, we prove the variance-reduction property:

$$\mathrm{Var}\left[\mu_{\mathrm{RB}}^{(N)}\right]$$

$$= \mathop{\mathbb{E}}_{\overline{\boldsymbol{Y}}'} \left[ \left( \sum_{n=1}^{N} \mathop{\mathbb{E}}_{\overline{\boldsymbol{Y}}} \left[ \mu_{\mathrm{MC}}^{n}(\overline{\boldsymbol{Y}}) \,\middle|\, \overline{\boldsymbol{Y}}_{<n} = \overline{\boldsymbol{Y}}'_{<n} \right] \right)^{2} \right] - \mathbb{E}[\mu_{\mathrm{MC}}^{(N)}]^{2} \qquad (35a)$$

(defintion of $\mu_{\mathrm{RB}}^{(N)}$ and unbiasedness)

$$= \mathop{\mathbb{E}}_{\overline{\boldsymbol{Y}}'} \left[ \left( \sum_{n=1}^{N} \mathop{\mathbb{E}}_{\overline{\boldsymbol{Y}}^{n}} \left[ \mu_{\mathrm{MC}}^{n}(\overline{\boldsymbol{Y}}^{n}) \,\middle|\, \overline{\boldsymbol{Y}}^{n}_{<n} = \overline{\boldsymbol{Y}}'_{<n} \right] \right)^{2} \right] - \mathbb{E}[\mu_{\mathrm{RB}}^{(N)}]^{2} \qquad (35b)$$

(each $\overline{\boldsymbol{Y}}^{n}$ is distributed independently and identically to $\overline{\boldsymbol{Y}}$)

$$= \mathop{\mathbb{E}}_{\overline{\boldsymbol{Y}}'} \left[ \left( \mathop{\mathbb{E}}_{\overline{\boldsymbol{Y}}^{1}} \left[ \cdots \mathop{\mathbb{E}}_{\overline{\boldsymbol{Y}}^{N}} \left[ \sum_{n=1}^{N} \mu_{\mathrm{MC}}^{n}(\overline{\boldsymbol{Y}}^{n}) \,\middle|\, \overline{\boldsymbol{Y}}^{N}_{<N} = \overline{\boldsymbol{Y}}'_{<N} \right] \cdots \,\middle|\, \overline{\boldsymbol{Y}}^{1}_{<1} = \overline{\boldsymbol{Y}}'_{<1} \right] \right)^{2} \right] + \mathbb{E}[\mu_{\mathrm{MC}}^{(N)}]^{2} \quad (35c)$$

(linearity of expectation)

$$\leq \mathop{\mathbb{E}}_{\overline{\boldsymbol{Y}}'} \left[ \mathop{\mathbb{E}}_{\overline{\boldsymbol{Y}}^{1}} \left[ \cdots \mathop{\mathbb{E}}_{\overline{\boldsymbol{Y}}^{N}} \left[ \left( \sum_{n=1}^{N} \mu_{\mathrm{MC}}^{n}(\overline{\boldsymbol{Y}}^{n}) \right)^{2} \,\middle|\, \overline{\boldsymbol{Y}}^{N}_{<N} = \overline{\boldsymbol{Y}}'_{<N} \right] \cdots \,\middle|\, \overline{\boldsymbol{Y}}^{1}_{<1} = \overline{\boldsymbol{Y}}'_{<1} \right] \right] - \mathbb{E}[\mu_{\mathrm{MC}}^{(N)}]^{2} \quad (35d)$$

(Jensen's inequality)

$$= \mathop{\mathbb{E}}_{\overline{\boldsymbol{Y}}^{1}} \left[ \cdots \mathop{\mathbb{E}}_{\overline{\boldsymbol{Y}}^{N}} \left[ \left( \sum_{n=1}^{N} \mu_{\mathrm{MC}}^{n}(\overline{\boldsymbol{Y}}^{n}) \right)^{2} \right] \right] - \mathbb{E}[\mu_{\mathrm{MC}}^{(N)}]^{2} \qquad (35e)$$

(law of total expectation)

$$= \mathop{\mathbb{E}}_{\overline{\boldsymbol{Y}}} \left[ \left( \sum_{n=1}^{N} \mu_{\mathrm{MC}}^{n}(\overline{\boldsymbol{Y}}) \right)^{2} \right] - \mathbb{E}[\mu_{\mathrm{MC}}^{(N)}]^{2} \qquad (35f)$$

$$(\{\overline{\boldsymbol{Y}}^n\}_{n=1}^N \text{ are i.i.d.})$$

$$= \underset{\overline{\boldsymbol{Y}}}{\mathbb{E}}\left[\left(\mu_{\text{MC}}^{(N)}(\overline{\boldsymbol{Y}})\right)^2\right] - \mathbb{E}[\mu_{\text{MC}}^{(N)}]^2 \tag{35g}$$

$$(\text{definition of } \mu_{\text{MC}}^{(N)})$$

$$= \text{Var}\left[\mu_{\text{MC}}^{(N)}\right] \tag{35h}$$

$$(\text{definition of variance})$$

■

**Lemma 12.**

$$\sum_{n=1}^{\infty} \mu_{\text{MC}}^n = \mu_{\text{MC}} \tag{36}$$

*Proof.*

$$\mu_{\text{MC}} = \frac{1}{M} \sum_{m=1}^{M} \log \frac{p(\boldsymbol{Y}^{(m)})}{q(\boldsymbol{Y}^{(m)})} \tag{37a}$$

$$= \frac{1}{M} \sum_{m=1}^{M} \log \frac{\vec{p}(\text{EOS} \mid \boldsymbol{Y}^{(m)})}{\vec{q}(\text{EOS} \mid \boldsymbol{Y}^{(m)})} \prod_{n=1}^{|\boldsymbol{Y}^{(m)}|} \log \frac{\vec{p}(Y_n^{(m)} \mid \boldsymbol{Y}_{<n}^{(m)})}{\vec{q}(Y_n^{(m)} \mid \boldsymbol{Y}_{<n}^{(m)})} \tag{37b}$$

$$= \frac{1}{M} \sum_{m=1}^{M} \log \prod_{n=1}^{\infty} \frac{\vec{p}(\overline{Y}_n^{(m)} \mid \overline{\boldsymbol{Y}}_{<n}^{(m)})}{\vec{q}(\overline{Y}_n^{(m)} \mid \overline{\boldsymbol{Y}}_{<n}^{(m)})} \tag{37c}$$

$$= \frac{1}{M} \sum_{m=1}^{M} \log \lim_{N \to \infty} \prod_{n=1}^{N} \frac{\vec{p}(\overline{Y}_n^{(m)} \mid \overline{\boldsymbol{Y}}_{<n}^{(m)})}{\vec{q}(\overline{Y}_n^{(m)} \mid \overline{\boldsymbol{Y}}_{<n}^{(m)})} \tag{37d}$$

$$= \lim_{N \to \infty} \frac{1}{M} \sum_{m=1}^{M} \log \prod_{n=1}^{N} \frac{\vec{p}(\overline{Y}_n^{(m)} \mid \overline{\boldsymbol{Y}}_{<n}^{(m)})}{\vec{q}(\overline{Y}_n^{(m)} \mid \overline{\boldsymbol{Y}}_{<n}^{(m)})} \tag{37e}$$

$$= \lim_{N \to \infty} \frac{1}{M} \sum_{m=1}^{M} \log \frac{\vec{p}(\overline{\boldsymbol{Y}}_{\leq N}^{(m)})}{\vec{q}(\overline{\boldsymbol{Y}}_{\leq N}^{(m)})} \tag{37f}$$

$$= \lim_{N \to \infty} \sum_{n=1}^{N} \mu_{\text{MC}}^n \tag{37g}$$

Note that going from Eq. (37b) to Eq. (37c), we use the padding construction given in §3. ■

**Theorem 2.** *Suppose the MC estimator $\mu_{\text{MC}}$ has finite variance, i.e., $\text{Var}[\mu_{\text{MC}}] < \infty$. Then the following properties regarding $\mu_{\text{RB}}$ hold:*

$(i)$ $\mathbb{E}[\mu_{\text{RB}}] = \text{KL}(p \parallel q)$  *(unbiasedness)*      $(ii)$ $\text{Var}[\mu_{\text{RB}}] \leq \text{Var}[\mu_{\text{MC}}]$  *(variance reduction)*

*Proof.* In this proof, we consider the special case of $M = 1$. The proof easily generalizes to $M > 1$ using the i.i.d. assumption. Additionally, we write $\mu_{\text{RB}}(\overline{\boldsymbol{Y}})$, rather than suppressing the argument to the estimator as we do in the main text. We begin with proving the unbiasedness of the estimator, using Lemma 12 and Lemma 11.

$$\mathbb{E}\left[\mu_{\text{RB}}(\overline{\boldsymbol{Y}})\right] = \underset{\overline{\boldsymbol{Y}}'}{\mathbb{E}}\left[\lim_{N \to \infty} \sum_{n=1}^{N} \underset{\overline{\boldsymbol{Y}}}{\mathbb{E}}\left[\mu_{\text{MC}}^n(\overline{\boldsymbol{Y}}) \mid \overline{\boldsymbol{Y}}_{<n} = \overline{\boldsymbol{Y}}'_{<n}\right]\right] \tag{38a}$$

$$= \lim_{N \to \infty} \sum_{n=1}^{N} \underset{\overline{\boldsymbol{Y}}'}{\mathbb{E}}\left[\underset{\overline{\boldsymbol{Y}}}{\mathbb{E}}\left[\mu_{\text{MC}}^n(\overline{\boldsymbol{Y}}) \mid \overline{\boldsymbol{Y}}_{<n} = \overline{\boldsymbol{Y}}'_{<n}\right]\right] \tag{38b}$$

$$(\text{Tonelli's Theorem})$$

$$= \lim_{N \to \infty} \sum_{n=1}^{N} \mathop{\mathbb{E}}_{\overline{\boldsymbol{Y}}} \left[ \mu_{\mathrm{MC}}^n(\overline{\boldsymbol{Y}}) \right] \tag{38c}$$

$$= \mathop{\mathbb{E}}_{\overline{\boldsymbol{Y}}} \left[ \lim_{N \to \infty} \sum_{n=1}^{N} \mu_{\mathrm{MC}}^n(\overline{\boldsymbol{Y}}) \right] \tag{38d}$$

(Fubini's Theorem, $\mathbb{E}[\sum_{n=1}^{\infty} |\mu_{\mathrm{MC}}^n(\overline{\boldsymbol{Y}})|] < \infty$)

$$= \mathop{\mathbb{E}}_{\overline{\boldsymbol{Y}}} \left[ \mu_{\mathrm{MC}}(\overline{\boldsymbol{Y}}) \right] \tag{38e}$$

(Lemma 12)

$$= \mathrm{KL}(p \,||\, q). \tag{38f}$$

(unbiasedness of $\mu_{\mathrm{MC}}$)

Finally, we prove the variance-reduction property.

$$\mathrm{Var}[\mu_{\mathrm{RB}}] = \mathop{\mathbb{E}}_{\overline{\boldsymbol{Y}}'} \left[ \left( \lim_{N \to \infty} \sum_{n=1}^{N} \mathop{\mathbb{E}}_{\overline{\boldsymbol{Y}}} \left[ \mu_{\mathrm{MC}}^n(\overline{\boldsymbol{Y}}) \mid \overline{\boldsymbol{Y}}_{<n} = \overline{\boldsymbol{Y}}'_{<n} \right] - \mathrm{KL}(p \,||\, q) \right)^2 \right] \tag{39a}$$

(Definition of $\mu_{\mathrm{RB}}(\boldsymbol{Y})$)

$$= \lim_{N \to \infty} \mathop{\mathbb{E}}_{\overline{\boldsymbol{Y}}'} \left[ \left( \sum_{n=1}^{N} \mathop{\mathbb{E}}_{\overline{\boldsymbol{Y}}} \left[ \mu_{\mathrm{MC}}^n(\overline{\boldsymbol{Y}}) \mid \overline{\boldsymbol{Y}}_{<n} = \overline{\boldsymbol{Y}}'_{<n} \right] - \mathrm{KL}(p \,||\, q) \right)^2 \right] \tag{39b}$$

(dominated convergence theorem, $\mathrm{Var}[\mu_{\mathrm{RB}}] < \infty$)

$$\leq \lim_{N \to \infty} \mathop{\mathbb{E}}_{\overline{\boldsymbol{Y}}} \left[ \left( \sum_{n=1}^{N} \mu_{\mathrm{MC}}^n(\overline{\boldsymbol{Y}}) - \mathrm{KL}(p, q) \right)^2 \right] \tag{39c}$$

(Lemma 11, variance reduction)

$$= \mathop{\mathbb{E}}_{\overline{\boldsymbol{Y}}} \left[ \lim_{N \to \infty} \left( \sum_{n=1}^{N} \mu_{\mathrm{MC}}^n(\overline{\boldsymbol{Y}}) - \mathrm{KL}(p, q) \right)^2 \right] \tag{39d}$$

(dominated convergence theorem, $\mathrm{Var}[\mu_{\mathrm{MC}}] < \infty$)

$$= \mathop{\mathbb{E}}_{\overline{\boldsymbol{Y}}} \left[ \left( \mu_{\mathrm{MC}}(\overline{\boldsymbol{Y}}) - \mathrm{KL}(p \,||\, q) \right)^2 \right] \tag{39e}$$

(Lemma 12)

$$= \mathrm{Var}[\mu_{\mathrm{MC}}]. \tag{39f}$$

∎

# D  Rao–Blackwellized Estimator of the Gradient

**Theorem 4.** *Let $p_{\boldsymbol{\theta}}$ and $q$ be two language models over $\Sigma$ and $\vec{p}_{\boldsymbol{\theta}}$ the prefix probability function of $p_{\boldsymbol{\theta}}$. Then, the following equality holds*

$$\nabla_{\boldsymbol{\theta}} \mathrm{KL}(p_{\boldsymbol{\theta}} \parallel q) = \sum_{\boldsymbol{y} \in \Sigma^*} \vec{p}_{\boldsymbol{\theta}}(\boldsymbol{y}) \, \mathbb{E}_{Y} \left[ \log \frac{\vec{p}_{\boldsymbol{\theta}}(Y \mid \boldsymbol{y})}{\vec{q}(Y \mid \boldsymbol{y})} \cdot (\nabla_{\boldsymbol{\theta}} \log \vec{p}_{\boldsymbol{\theta}}(Y \mid \boldsymbol{y}) + \nabla_{\boldsymbol{\theta}} \log \vec{p}_{\boldsymbol{\theta}}(\boldsymbol{y})) \right]. \quad (15)$$

*Proof.*

$$\nabla_{\boldsymbol{\theta}} \mathrm{KL}(p \parallel q)$$

$$= \sum_{\boldsymbol{y} \in \Sigma^*} \mathrm{KL}(\vec{p}_{\boldsymbol{\theta}}(\cdot \mid \boldsymbol{y}) \parallel \vec{q}(\cdot \mid \boldsymbol{y})) \nabla_{\boldsymbol{\theta}} \vec{p}_{\boldsymbol{\theta}}(\boldsymbol{y}) + \vec{p}_{\boldsymbol{\theta}}(\boldsymbol{y}) \nabla_{\boldsymbol{\theta}} \mathrm{KL}(\vec{p}_{\boldsymbol{\theta}}(\cdot \mid \boldsymbol{y}) \parallel \vec{q}(\cdot \mid \boldsymbol{y})) \quad (40a)$$

$$= \sum_{\boldsymbol{y} \in \Sigma^*} \mathrm{KL}(\vec{p}_{\boldsymbol{\theta}}(\cdot \mid \boldsymbol{y}) \parallel \vec{q}(\cdot \mid \boldsymbol{y})) \nabla_{\boldsymbol{\theta}} \vec{p}_{\boldsymbol{\theta}}(\boldsymbol{y}) + \vec{p}_{\boldsymbol{\theta}}(\boldsymbol{y}) \sum_{\overline{y} \in \Sigma} \nabla_{\boldsymbol{\theta}} \vec{p}_{\boldsymbol{\theta}}(\overline{y} \mid \boldsymbol{y}) \log \frac{\vec{p}_{\boldsymbol{\theta}}(\overline{y} \mid \boldsymbol{y})}{\vec{q}(\overline{y} \mid \boldsymbol{y})} \quad (40b)$$

$$= \sum_{\boldsymbol{y} \in \Sigma^*} \mathrm{KL}(\vec{p}_{\boldsymbol{\theta}}(\cdot \mid \boldsymbol{y}) \parallel \vec{q}(\cdot \mid \boldsymbol{y})) \nabla_{\boldsymbol{\theta}} \vec{p}_{\boldsymbol{\theta}}(\boldsymbol{y}) + \vec{p}_{\boldsymbol{\theta}}(\boldsymbol{y}) \sum_{\overline{y} \in \Sigma} \log \frac{\vec{p}_{\boldsymbol{\theta}}(\overline{y} \mid \boldsymbol{y})}{\vec{q}(\overline{y} \mid \boldsymbol{y})} \nabla_{\boldsymbol{\theta}} \vec{p}_{\boldsymbol{\theta}}(\overline{y} \mid \boldsymbol{y}) + \nabla_{\boldsymbol{\theta}} \vec{p}_{\boldsymbol{\theta}}(\overline{y} \mid \boldsymbol{y})$$

$$\quad (40c)$$

$$= \sum_{\boldsymbol{y} \in \Sigma^*} \mathrm{KL}(\vec{p}_{\boldsymbol{\theta}}(\cdot \mid \boldsymbol{y}) \parallel \vec{q}(\cdot \mid \boldsymbol{y})) \nabla_{\boldsymbol{\theta}} \vec{p}_{\boldsymbol{\theta}}(\boldsymbol{y}) + \vec{p}_{\boldsymbol{\theta}}(\boldsymbol{y}) \, \mathbb{E}_{\overline{Y}} \left[ \left( \log \frac{\vec{p}_{\boldsymbol{\theta}}(\overline{Y} \mid \boldsymbol{y})}{\vec{q}(\overline{Y} \mid \boldsymbol{y})} + 1 \right) \nabla_{\boldsymbol{\theta}} \log \vec{p}_{\boldsymbol{\theta}}(\overline{Y} \mid \boldsymbol{y}) \right]$$

$$\quad (40d)$$

$$= \sum_{\boldsymbol{y} \in \Sigma^*} \mathrm{KL}(\vec{p}_{\boldsymbol{\theta}}(\cdot \mid \boldsymbol{y}) \parallel \vec{q}(\cdot \mid \boldsymbol{y})) \nabla_{\boldsymbol{\theta}} \vec{p}_{\boldsymbol{\theta}}(\boldsymbol{y}) + \vec{p}_{\boldsymbol{\theta}}(\boldsymbol{y}) \, \mathbb{E}_{\overline{Y}} \left[ \log \frac{\vec{p}_{\boldsymbol{\theta}}(\overline{Y} \mid \boldsymbol{y})}{\vec{q}(\overline{Y} \mid \boldsymbol{y})} \nabla_{\boldsymbol{\theta}} \log \vec{p}_{\boldsymbol{\theta}}(\overline{Y} \mid \boldsymbol{y}) \right] \quad (40e)$$

$$= \sum_{\boldsymbol{y} \in \Sigma^*} \mathbb{E}_{\overline{Y}} \left[ \log \frac{\vec{p}_{\boldsymbol{\theta}}(\overline{Y} \mid \boldsymbol{y})}{\vec{q}(\overline{Y} \mid \boldsymbol{y})} \right] \nabla_{\boldsymbol{\theta}} \vec{p}_{\boldsymbol{\theta}}(\boldsymbol{y}) + \vec{p}_{\boldsymbol{\theta}}(\boldsymbol{y}) \, \mathbb{E}_{\overline{Y}} \left[ \log \frac{\vec{p}_{\boldsymbol{\theta}}(\overline{Y} \mid \boldsymbol{y})}{\vec{q}(\overline{Y} \mid \boldsymbol{y})} \nabla_{\boldsymbol{\theta}} \log \vec{p}_{\boldsymbol{\theta}}(\overline{Y} \mid \boldsymbol{y}) \right] \quad (40f)$$

$$= \sum_{\boldsymbol{y} \in \Sigma^*} \vec{p}_{\boldsymbol{\theta}}(\boldsymbol{y}) \, \mathbb{E}_{\overline{Y}} \left[ \log \frac{\vec{p}_{\boldsymbol{\theta}}(\overline{Y} \mid \boldsymbol{y})}{\vec{q}(\overline{Y} \mid \boldsymbol{y})} \right] \nabla_{\boldsymbol{\theta}} \log \vec{p}_{\boldsymbol{\theta}}(\boldsymbol{y}) + \vec{p}_{\boldsymbol{\theta}}(\boldsymbol{y}) \, \mathbb{E}_{\overline{Y}} \left[ \log \frac{\vec{p}_{\boldsymbol{\theta}}(\overline{Y} \mid \boldsymbol{y})}{\vec{q}(\overline{Y} \mid \boldsymbol{y})} \nabla_{\boldsymbol{\theta}} \log \vec{p}_{\boldsymbol{\theta}}(\overline{Y} \mid \boldsymbol{y}) \right]$$

$$\quad (40g)$$

$$= \sum_{\boldsymbol{y} \in \Sigma^*} \vec{p}_{\boldsymbol{\theta}}(\boldsymbol{y}) \, \mathbb{E}_{\overline{Y}} \left[ \log \frac{\vec{p}_{\boldsymbol{\theta}}(\overline{Y} \mid \boldsymbol{y})}{\vec{q}(\overline{Y} \mid \boldsymbol{y})} \left( \nabla_{\boldsymbol{\theta}} \log \vec{p}_{\boldsymbol{\theta}}(\overline{Y} \mid \boldsymbol{y}) + \nabla_{\boldsymbol{\theta}} \log \vec{p}_{\boldsymbol{\theta}}(\boldsymbol{y}) \right) \right]. \quad (40h)$$

∎

## D.1  Variance-Reduction Proof

The proof structure is as follows: We first prove that the inequality holds when we constrain $\overline{Y}$ to have length less than or equal to $N$. We then generalize to the infinite-length sequences by analyzing as $N \to \infty$. We begin with defining the truncated MC and RB estimators. Let $\boldsymbol{\delta}_{\mathrm{MC}}^{(N)}$ be the truncated MC estimator of the gradient:

$$\boldsymbol{\delta}_{\mathrm{MC}}^{(N)} = \sum_{n=1}^{N} \frac{1}{M} \sum_{m=1}^{M} \log \frac{\vec{p}_{\boldsymbol{\theta}}(\overline{Y}_n^{(m)} \mid \overline{\boldsymbol{Y}}_{<n}^{(m)})}{\vec{q}(\overline{Y}_n^{(m)} \mid \overline{\boldsymbol{Y}}_{<n}^{(m)})} \nabla_{\boldsymbol{\theta}} \log \vec{p}_{\boldsymbol{\theta}}(\overline{\boldsymbol{Y}}_{\leq N}^{(m)}). \quad (41)$$

Let $\boldsymbol{\delta}_{\mathrm{RB}}^{(N)}$ be the truncated RB estimator:

$$\boldsymbol{\delta}_{\mathrm{RB}}^{(N)} = \frac{1}{M} \sum_{m=1}^{M} \sum_{n=1}^{N} \mathbb{E}_{\overline{Y}_n} \left[ \log \frac{\vec{p}_{\boldsymbol{\theta}}(\overline{Y}_n \mid \overline{\boldsymbol{Y}}_{<n}^{(m)})}{\vec{q}(\overline{Y}_n \mid \overline{\boldsymbol{Y}}_{<n}^{(m)})} \left( \nabla_{\boldsymbol{\theta}} \log \vec{p}_{\boldsymbol{\theta}}(\overline{Y}_n \mid \overline{\boldsymbol{Y}}_{<n}^{(m)}) + \nabla_{\boldsymbol{\theta}} \log \vec{p}_{\boldsymbol{\theta}}(\overline{\boldsymbol{Y}}_{<n}^{(m)}) \right) \right]. $$

$$\quad (42)$$

**Lemma 13.** *The truncated MC estimator of the gradient $\boldsymbol{\delta}_{\mathrm{MC}}^{(N)}$ converges to $\boldsymbol{\delta}_{\mathrm{MC}}$ as $N$ goes to $\infty$, i.e.,* $\lim_{N\to\infty}\boldsymbol{\delta}_{\mathrm{MC}}^{(N)} = \boldsymbol{\delta}_{\mathrm{MC}}$.

*Proof.*

$$\boldsymbol{\delta}_{\mathrm{MC}} = \frac{1}{M}\sum_{m=1}^{M}\log\frac{\vec{p}_{\boldsymbol{\theta}}(\boldsymbol{Y}^{(m)})}{\vec{q}(\boldsymbol{Y}^{(m)})}\nabla_{\boldsymbol{\theta}}\log\vec{p}_{\boldsymbol{\theta}}(\boldsymbol{Y}^{(m)}) \tag{43}$$

$$= \frac{1}{M}\sum_{m=1}^{M}\log\frac{\vec{p}_{\boldsymbol{\theta}}(\mathrm{EOS}\mid\boldsymbol{Y}^{(m)})}{\vec{q}(\mathrm{EOS}\mid\boldsymbol{Y}^{(m)})}\prod_{n=1}^{|\boldsymbol{Y}^{(m)}|}\frac{\vec{p}_{\boldsymbol{\theta}}(Y_n\mid\boldsymbol{Y}_{<n}^{(m)})}{\vec{q}(Y_n\mid\boldsymbol{Y}_{<n}^{(m)})}\nabla_{\boldsymbol{\theta}}\log\vec{p}_{\boldsymbol{\theta}}(\boldsymbol{Y}^{(m)}) \tag{44}$$

$$= \frac{1}{M}\sum_{m=1}^{M}\log\prod_{n=1}^{\infty}\frac{\vec{p}_{\boldsymbol{\theta}}(\overline{Y}_n\mid\overline{\boldsymbol{Y}}_{<n}^{(m)})}{\vec{q}(\overline{Y}_n\mid\overline{\boldsymbol{Y}}_{<n}^{(m)})}\nabla_{\boldsymbol{\theta}}\log\vec{p}_{\boldsymbol{\theta}}(\overline{\boldsymbol{Y}}^{(m)}) \tag{45}$$

$$= \frac{1}{M}\sum_{m=1}^{M}\log\lim_{N\to\infty}\prod_{n=1}^{N}\frac{\vec{p}_{\boldsymbol{\theta}}(\overline{Y}_n\mid\overline{\boldsymbol{Y}}_{<n}^{(m)})}{\vec{q}(\overline{Y}_n\mid\overline{\boldsymbol{Y}}_{<n}^{(m)})}\nabla_{\boldsymbol{\theta}}\log\vec{p}_{\boldsymbol{\theta}}(\overline{\boldsymbol{Y}}_{\leq N}^{(m)}) \tag{46}$$

$$= \frac{1}{M}\sum_{m=1}^{M}\lim_{N\to\infty}\log\prod_{n=1}^{N}\frac{\vec{p}_{\boldsymbol{\theta}}(\overline{Y}_n\mid\overline{\boldsymbol{Y}}_{<n}^{(m)})}{\vec{q}(\overline{Y}_n\mid\overline{\boldsymbol{Y}}_{<n}^{(m)})}\nabla_{\boldsymbol{\theta}}\log\vec{p}_{\boldsymbol{\theta}}(\overline{\boldsymbol{Y}}_{\leq N}^{(m)}) \tag{47}$$

$$= \lim_{N\to\infty}\frac{1}{M}\sum_{m=1}^{M}\log\frac{\vec{p}_{\boldsymbol{\theta}}(\overline{\boldsymbol{Y}}_{\leq N}^{(m)})}{\vec{q}(\overline{\boldsymbol{Y}}_{\leq N}^{(m)})}\nabla_{\boldsymbol{\theta}}\log\vec{p}_{\boldsymbol{\theta}}(\overline{\boldsymbol{Y}}_{\leq N}^{(m)}) \tag{48}$$

$$= \lim_{N\to\infty}\boldsymbol{\delta}_{\mathrm{MC}}^{(N)}. \tag{49}$$

∎

**Lemma 14.** *The following identity holds:*

$$\nabla_{\boldsymbol{\theta}}\log\vec{p}_{\boldsymbol{\theta}}(\overline{\boldsymbol{Y}}_{\leq n}) = \underset{\overline{\boldsymbol{Y}}'}{\mathbb{E}}\left[\nabla_{\boldsymbol{\theta}}\log\vec{p}_{\boldsymbol{\theta}}(\overline{\boldsymbol{Y}}'_{\leq N})\,\Big|\,\overline{\boldsymbol{Y}}'_{\leq n} = \overline{\boldsymbol{Y}}_{\leq n}\right], \tag{50}$$

*for any $N > n$.*

*Proof.* Note that $\vec{p}_{\boldsymbol{\theta}}(\overline{\boldsymbol{Y}}_{\leq n}) = \sum_{\overline{\boldsymbol{y}}'\in\overline{\Sigma}^{N-n}}\vec{p}_{\boldsymbol{\theta}}(\overline{\boldsymbol{Y}}_{\leq n}\overline{\boldsymbol{y}}')$ for any $N > n$. Therefore, we have

$$\nabla_{\boldsymbol{\theta}}\log\vec{p}_{\boldsymbol{\theta}}(\overline{\boldsymbol{Y}}_{\leq n}) = \frac{\nabla_{\boldsymbol{\theta}}\vec{p}_{\boldsymbol{\theta}}(\overline{\boldsymbol{Y}}_{\leq n})}{\vec{p}_{\boldsymbol{\theta}}(\overline{\boldsymbol{Y}}_{\leq n})} \tag{51a}$$

$$= \sum_{\overline{\boldsymbol{y}}'\in\overline{\Sigma}^{N-n}}\frac{\nabla_{\boldsymbol{\theta}}\vec{p}_{\boldsymbol{\theta}}(\overline{\boldsymbol{Y}}_{\leq n}\overline{\boldsymbol{y}}')}{\vec{p}_{\boldsymbol{\theta}}(\overline{\boldsymbol{Y}}_{\leq n})} \tag{51b}$$

$$= \sum_{\overline{\boldsymbol{y}}'\in\overline{\Sigma}^{N-n}}\frac{\vec{p}_{\boldsymbol{\theta}}(\overline{\boldsymbol{Y}}_{\leq n}\overline{\boldsymbol{y}}')}{\vec{p}_{\boldsymbol{\theta}}(\overline{\boldsymbol{Y}}_{\leq n})}\nabla_{\boldsymbol{\theta}}\log\vec{p}_{\boldsymbol{\theta}}(\overline{\boldsymbol{Y}}_{\leq n}\overline{\boldsymbol{y}}') \tag{51c}$$

$$= \sum_{\overline{\boldsymbol{y}}'\in\overline{\Sigma}^{N-n}}\frac{\vec{p}_{\boldsymbol{\theta}}(\overline{\boldsymbol{y}}'\mid\overline{\boldsymbol{Y}}_{\leq n})\vec{p}_{\boldsymbol{\theta}}(\overline{\boldsymbol{Y}}_{\leq n})}{\vec{p}_{\boldsymbol{\theta}}(\overline{\boldsymbol{Y}}_{\leq n})}\nabla_{\boldsymbol{\theta}}\log\vec{p}_{\boldsymbol{\theta}}(\overline{\boldsymbol{Y}}_{\leq n}\overline{\boldsymbol{y}}') \tag{51d}$$

$$= \sum_{\overline{\boldsymbol{y}}'\in\overline{\Sigma}^{N-n}}\vec{p}_{\boldsymbol{\theta}}(\overline{\boldsymbol{y}}'\mid\overline{\boldsymbol{Y}}_{\leq n})\nabla_{\boldsymbol{\theta}}\log\vec{p}_{\boldsymbol{\theta}}(\overline{\boldsymbol{Y}}_{\leq n}\overline{\boldsymbol{y}}') \tag{51e}$$

$$= \underset{\overline{\boldsymbol{Y}}'}{\mathbb{E}}\left[\nabla_{\boldsymbol{\theta}}\log\vec{p}_{\boldsymbol{\theta}}(\overline{\boldsymbol{Y}}'_{\leq N})\,\Big|\,\overline{\boldsymbol{Y}}'_{\leq n} = \overline{\boldsymbol{Y}}_{\leq n}\right] \tag{51f}$$

∎

**Lemma 15.** *Let $\boldsymbol{\delta}_{\mathrm{MC}}^{(N)}$ be the truncated MC estimator of the gradient defined in Eq. (41) and $\boldsymbol{\delta}_{\mathrm{RB}}^{(N)}$ the truncated RB estimator of the gradient defined in Eq. (42). Define $\boldsymbol{G}^N \overset{\mathrm{def}}{=} \mathbb{E}[\boldsymbol{\delta}_{\mathrm{MC}}^{(N)}] = \mathbb{E}[\boldsymbol{\delta}_{\mathrm{RB}}^{(N)}]$. We have*

$$\mathbb{E}\left[\left\|\boldsymbol{\delta}_{\mathrm{RB}}^{(N)} - \boldsymbol{G}^N\right\|^2\right] \leq \mathbb{E}\left[\left\|\boldsymbol{\delta}_{\mathrm{MC}}^{(N)} - \boldsymbol{G}^N\right\|^2\right]. \tag{52}$$

*Proof.* Without loss of generality, we assume $M = 1$. The proof generalizes to $M > 1$ with the i.i.d. assumption.

$$\mathbb{E}\left[\left\|\boldsymbol{\delta}_{\mathrm{RB}}^{(N)} - \boldsymbol{G}^N\right\|^2\right] \tag{53a}$$

$$= \mathbb{E}_{\overline{\boldsymbol{Y}}'}\left[\left\|\sum_{n=1}^N \mathbb{E}_{\overline{\boldsymbol{Y}}}\left[\mu_{\mathrm{MC}}^n(\overline{\boldsymbol{Y}}_{\leq n})\underbrace{\left(\nabla_{\boldsymbol{\theta}}\log\vec{p}_{\boldsymbol{\theta}}(\overline{\boldsymbol{Y}}_{<n}) + \nabla_{\boldsymbol{\theta}}\log\vec{p}_{\boldsymbol{\theta}}(\overline{Y}_n \mid \overline{\boldsymbol{Y}}_{<n})\right)}_{\nabla_{\boldsymbol{\theta}}\log\vec{p}_{\boldsymbol{\theta}}(\overline{\boldsymbol{Y}}_{\leq n})} - \boldsymbol{G}^N \,\Big|\, \overline{\boldsymbol{Y}}_{<n} = \overline{\boldsymbol{Y}}'_{<n}\right]\right\|^2\right] \tag{53b}$$

(definition of $\boldsymbol{\delta}_{\mathrm{RB}}^{(N)}$)

$$= \mathbb{E}_{\overline{\boldsymbol{Y}}'}\left[\left\|\sum_{n=1}^N \mathbb{E}_{\overline{\boldsymbol{Y}}^n}\left[\mu_{\mathrm{MC}}^n(\overline{\boldsymbol{Y}}_{\leq n}^n)\nabla_{\boldsymbol{\theta}}\log\vec{p}_{\boldsymbol{\theta}}(\overline{\boldsymbol{Y}}_{\leq n}^n) - \boldsymbol{G}^N \,\Big|\, \overline{\boldsymbol{Y}}_{<n}^n = \overline{\boldsymbol{Y}}'_{<n}\right]\right\|^2\right] \tag{53c}$$

(each $\overline{\boldsymbol{Y}}^n$ is distributed independently and identically to $\overline{\boldsymbol{Y}}$)

$$= \mathbb{E}_{\overline{\boldsymbol{Y}}'}\left[\left\|\mathbb{E}_{\overline{\boldsymbol{Y}}^1}\left[\cdots \mathbb{E}_{\overline{\boldsymbol{Y}}^N}\left[\sum_{n=1}^N \mu_{\mathrm{MC}}^n(\overline{\boldsymbol{Y}}_{\leq n}^n)\nabla_{\boldsymbol{\theta}}\log\vec{p}_{\boldsymbol{\theta}}(\overline{\boldsymbol{Y}}_{\leq n}^n) - \boldsymbol{G}^N \,\Big|\, \overline{\boldsymbol{Y}}_{<N}^N = \overline{\boldsymbol{Y}}'_{<N}\right]\cdots \,\Big|\, \overline{\boldsymbol{Y}}_{<1}^1 = \overline{\boldsymbol{Y}}'_{<1}\right]\right\|^2\right] \tag{53d}$$

(linearity of expectation)

$$\leq \mathbb{E}_{\overline{\boldsymbol{Y}}'}\left[\mathbb{E}_{\overline{\boldsymbol{Y}}^1}\left[\cdots \mathbb{E}_{\overline{\boldsymbol{Y}}^N}\left[\left\|\sum_{n=1}^N \mu_{\mathrm{MC}}^n(\overline{\boldsymbol{Y}}_{\leq n}^n)\nabla_{\boldsymbol{\theta}}\log\vec{p}_{\boldsymbol{\theta}}(\overline{\boldsymbol{Y}}_{\leq n}^n) - \boldsymbol{G}^N\right\|^2 \,\Big|\, \overline{\boldsymbol{Y}}_{<N}^N = \overline{\boldsymbol{Y}}'_{<N}\right]\cdots \,\Big|\, \overline{\boldsymbol{Y}}_{<1}^1 = \overline{\boldsymbol{Y}}'_{<1}\right]\right] \tag{53e}$$

(Jensen's inequality)

$$= \mathbb{E}_{\overline{\boldsymbol{Y}}^1}\left[\cdots \mathbb{E}_{\overline{\boldsymbol{Y}}^N}\left[\left\|\sum_{n=1}^N \mu_{\mathrm{MC}}^n(\overline{\boldsymbol{Y}}_{\leq n}^n)\nabla_{\boldsymbol{\theta}}\log\vec{p}_{\boldsymbol{\theta}}(\overline{\boldsymbol{Y}}_{\leq n}^n) - \boldsymbol{G}^N\right\|^2\right]\right] \tag{53f}$$

(law of total expectation)

$$= \mathbb{E}_{\overline{\boldsymbol{Y}}}\left[\left\|\sum_{n=1}^N \mu_{\mathrm{MC}}^n(\overline{\boldsymbol{Y}}_{\leq n})\nabla_{\boldsymbol{\theta}}\log\vec{p}_{\boldsymbol{\theta}}(\overline{\boldsymbol{Y}}_{\leq n}) - \boldsymbol{G}^N\right\|^2\right] \tag{53g}$$

($\overline{\boldsymbol{Y}}^1, \ldots, \overline{\boldsymbol{Y}}^N$ are i.i.d.)

$$\leq \mathbb{E}_{\overline{\boldsymbol{Y}}}\left[\left\|\sum_{n=1}^N \mu_{\mathrm{MC}}^n(\overline{\boldsymbol{Y}}_{\leq n})\,\mathbb{E}_{\overline{\boldsymbol{Y}}'}\left[\nabla_{\boldsymbol{\theta}}\log\vec{p}_{\boldsymbol{\theta}}(\overline{\boldsymbol{Y}}'_{\leq N}) \mid \overline{\boldsymbol{Y}}'_{\leq n} = \overline{\boldsymbol{Y}}_{\leq n}\right] - \boldsymbol{G}^N\right\|^2\right] \tag{53h}$$

(Lemma 14)

$$= \mathbb{E}_{\overline{\boldsymbol{Y}}}\left[\left\|\sum_{n=1}^N \mathbb{E}_{\overline{\boldsymbol{Y}}'}\left[\mu_{\mathrm{MC}}^n(\overline{\boldsymbol{Y}}'_{\leq n})\nabla_{\boldsymbol{\theta}}\log\vec{p}_{\boldsymbol{\theta}}(\overline{\boldsymbol{Y}}'_{\leq N}) - \boldsymbol{G}^N \,\Big|\, \overline{\boldsymbol{Y}}'_{\leq n} = \overline{\boldsymbol{Y}}_{\leq n}\right]\right\|^2\right] \tag{53i}$$

(linearity of expectation)

$$= \mathbb{E}_{\overline{\boldsymbol{Y}}}\left[\left\|\mathbb{E}_{\overline{\boldsymbol{Y}}'^1}\left[\cdots \mathbb{E}_{\overline{\boldsymbol{Y}}'^N}\left[\sum_{n=1}^N \mu_{\mathrm{MC}}^n(\overline{\boldsymbol{Y}}_{\leq n}'^n)\nabla_{\boldsymbol{\theta}}\log\vec{p}_{\boldsymbol{\theta}}(\overline{\boldsymbol{Y}}_{\leq N}'^n) - \boldsymbol{G}^N \,\Big|\, \boldsymbol{Y}_{\leq N}'^N = \boldsymbol{Y}_{\leq N}\right]\cdots \,\Big|\, \boldsymbol{Y}_{\leq 1}'^1 = \overline{\boldsymbol{Y}}_{\leq 1}\right]\right\|^2\right] \tag{53j}$$

(linearity of expectation)

$$\leq \underset{\overline{\boldsymbol{Y}}}{\mathbb{E}}\left[\underset{\overline{\boldsymbol{Y}}'^1}{\mathbb{E}}\left[\cdots \underset{\overline{\boldsymbol{Y}}'^N}{\mathbb{E}}\left[\left\|\sum_{n=1}^{N}\mu_{\mathrm{MC}}^n(\overline{\boldsymbol{Y}}_{\leq n}'^n)\nabla_{\boldsymbol{\theta}}\log \vec{p}_{\boldsymbol{\theta}}(\overline{\boldsymbol{Y}}_{\leq N}'^n)-\boldsymbol{G}^N\right\|^2 \Bigg| \overline{\boldsymbol{Y}}_{\leq N}'^N=\overline{\boldsymbol{Y}}_{\leq N}\right]\cdots \Bigg| \overline{\boldsymbol{Y}}_{\leq 1}'^1=\overline{\boldsymbol{Y}}_{\leq 1}\right]\right]$$

(53k)

(Jensen's inequality)

$$= \underset{\overline{\boldsymbol{Y}}'^1}{\mathbb{E}}\left[\cdots \underset{\overline{\boldsymbol{Y}}'^N}{\mathbb{E}}\left[\left\|\sum_{n=1}^{N}\mu_{\mathrm{MC}}^n(\overline{\boldsymbol{Y}}_{\leq n}'^n)\nabla_{\boldsymbol{\theta}}\log \vec{p}_{\boldsymbol{\theta}}(\overline{\boldsymbol{Y}}_{\leq N}'^n)-\boldsymbol{G}^N\right\|^2\right]\right]$$

(53l)

(law of total expectation)

$$= \underset{\overline{\boldsymbol{Y}}'}{\mathbb{E}}\left[\left\|\sum_{n=1}^{N}\mu_{\mathrm{MC}}^n(\overline{\boldsymbol{Y}}_{\leq n}')\nabla_{\boldsymbol{\theta}}\log \vec{p}_{\boldsymbol{\theta}}(\overline{\boldsymbol{Y}}_{\leq N}')-\boldsymbol{G}^N\right\|^2\right]$$

(53m)

$(\overline{\boldsymbol{Y}}'^1,\cdots,\overline{\boldsymbol{Y}}'^N$ are i.i.d.)

$$= \underset{\overline{\boldsymbol{Y}}}{\mathbb{E}}\left[\left\|\boldsymbol{\delta}_{\mathrm{MC}}^{(N)}(\overline{\boldsymbol{Y}})-\boldsymbol{G}^N\right\|^2\right].$$

(53n)

(definition of $\boldsymbol{\delta}_{\mathrm{MC}}^{(N)}$)

■

**Theorem 5.** *Assuming* $\mathrm{Var}[\boldsymbol{\delta}_{\mathrm{RB}}]<\infty, \mathrm{Var}[\boldsymbol{\delta}_{\mathrm{MC}}]<\infty$, *we have*

$$\mathbb{E}\left[\|\boldsymbol{\delta}_{\mathrm{RB}}-\boldsymbol{G}\|^2\right]\leq \mathbb{E}\left[\|\boldsymbol{\delta}_{\mathrm{MC}}-\boldsymbol{G}\|^2\right].$$

(17)

*Proof.* Without loss of generality, we assume $M=1$. The proof generalizes to $M>1$ with the i.i.d. assumption.

$$\mathbb{E}\left[\|\boldsymbol{\delta}_{\mathrm{RB}}-\boldsymbol{G}\|^2\right]=\mathbb{E}\left[\left\|\lim_{N\to\infty}\boldsymbol{\delta}_{\mathrm{RB}}^{(N)}-\boldsymbol{G}^N\right\|^2\right]$$

(54a)

(definition of $\boldsymbol{\delta}_{\mathrm{RB}}$)

$$= \lim_{N\to\infty}\mathbb{E}\left[\left\|\boldsymbol{\delta}_{\mathrm{RB}}^{(N)}-\boldsymbol{G}^N\right\|^2\right]$$

(54b)

(dominated convergence theorem, $\mathrm{Var}[\boldsymbol{\delta}_{\mathrm{RB}}]<\infty$)

$$\leq \lim_{N\to\infty}\mathbb{E}\left[\left\|\boldsymbol{\delta}_{\mathrm{MC}}^{(N)}-\boldsymbol{G}^N\right\|^2\right]$$

(54c)

(Lemma 15)

$$= \mathbb{E}\left[\left\|\lim_{N\to\infty}\boldsymbol{\delta}_{\mathrm{MC}}^{(N)}-\boldsymbol{G}^N\right\|^2\right]$$

(54d)

(dominated convergence theorem, $\mathrm{Var}[\boldsymbol{\delta}_{\mathrm{MC}}]<\infty$)

$$= \mathbb{E}\left[\|\boldsymbol{\delta}_{\mathrm{MC}}-\boldsymbol{G}\|^2\right].$$

(54e)

(Lemma 13)

■

## D.2 A Note on Rao–Blackwellizing KL in Trust-Region Algorithms

The conventional Monte Carlo estimator of $\mathrm{KL}(p_{\boldsymbol{\theta}}\|q)$ used in the PPO algorithm in open-sourced RLHF libraries, e.g., [13, 40], is as follows:

$$\mu_{\mathrm{MC}}^{\mathrm{PPO}}=\frac{1}{M}\sum_{m=1}^{M}\frac{p_{\boldsymbol{\theta}}(\boldsymbol{Y}^{(m)})}{p_{\boldsymbol{\theta}_{\mathrm{old}}}(\boldsymbol{Y}^{(m)})}\log \frac{p_{\boldsymbol{\theta}_{\mathrm{old}}}(\boldsymbol{Y}^{(m)})}{q(\boldsymbol{Y}^{(m)})},$$

(55)

where $\boldsymbol{Y}^{(1)}, \cdots, \boldsymbol{Y}^{(M)} \overset{\text{i.i.d.}}{\sim} p_{\boldsymbol{\theta}_{\text{old}}}$. Notably, the expected value of the above estimator is *not equal to* $\text{KL}(p_{\boldsymbol{\theta}} \| q)$ and is

$$\mathbb{E}[\mu_{\text{MC}}^{\text{PPO}}] = \underset{\boldsymbol{Y} \sim p_{\boldsymbol{\theta}_{\text{old}}}}{\mathbb{E}} \left[ \frac{p_{\boldsymbol{\theta}}(\boldsymbol{Y})}{p_{\boldsymbol{\theta}_{\text{old}}}(\boldsymbol{Y})} \log \frac{p_{\boldsymbol{\theta}_{\text{old}}}(\boldsymbol{Y})}{q(\boldsymbol{Y})} \right] = \underset{\boldsymbol{Y} \sim p_{\boldsymbol{\theta}}}{\mathbb{E}} \left[ \log \frac{p_{\boldsymbol{\theta}_{\text{old}}}(\boldsymbol{Y})}{q(\boldsymbol{Y})} \right]. \tag{56}$$

A natural question at this point is: what is the relationship between $\mu_{\text{MC}}^{\text{PPO}}$ and $\mu_{\text{MC}}$, and why is minimizing $\mu_{\text{MC}}^{\text{PPO}}$ a valid proxy for minimizing $\mu_{\text{MC}}$? Crucially, the KL divergence between $p_{\boldsymbol{\theta}}$ and $q$ can be decomposed into the sum of $\mathbb{E}[\mu_{\text{MC}}^{\text{PPO}}]$ and the KL divergence between $p_{\boldsymbol{\theta}}$ and $p_{\boldsymbol{\theta}_{\text{old}}}$, as shown in the following equation:

$$\underbrace{\underset{\boldsymbol{Y} \sim p_{\boldsymbol{\theta}}}{\mathbb{E}} \left[ \log \frac{p_{\boldsymbol{\theta}_{\text{old}}}(\boldsymbol{Y})}{q(\boldsymbol{Y})} \right]}_{\mathbb{E}[\mu_{\text{MC}}^{\text{PPO}}]} + \underbrace{\underset{\boldsymbol{Y} \sim p_{\boldsymbol{\theta}}}{\mathbb{E}} \left[ \log \frac{p_{\boldsymbol{\theta}}(\boldsymbol{Y})}{p_{\boldsymbol{\theta}_{\text{old}}}(\boldsymbol{Y})} \right]}_{\text{trust region}, \text{KL}(p_{\boldsymbol{\theta}}, p_{\boldsymbol{\theta}_{\text{old}}})} = \underset{\boldsymbol{Y} \sim p_{\boldsymbol{\theta}}}{\mathbb{E}} \left[ \log \frac{p_{\boldsymbol{\theta}}(\boldsymbol{Y})}{q(\boldsymbol{Y})} \right] = \text{KL}(p_{\boldsymbol{\theta}} \| q). \tag{57}$$

Therefore, minimizing $\text{KL}(p_{\boldsymbol{\theta}} \| q)$ is equivalent to minimizing both $\mathbb{E}[\mu_{\text{MC}}^{\text{PPO}}]$ and $\text{KL}(p_{\boldsymbol{\theta}} \| p_{\boldsymbol{\theta}_{\text{old}}})$. Notably, since the KL divergence between the current policy and the old policy, $\text{KL}(p_{\boldsymbol{\theta}} \| p_{\boldsymbol{\theta}_{\text{old}}})$, is already constrained by PPO's clipping mechanism, the algorithm effectively focuses on penalizing only the first term, using $\mu_{\text{MC}}^{\text{PPO}}$.

A naïve approach to Rao–Blackwellizing $\mu_{\text{MC}}^{\text{PPO}}$ defined in Eq. (55), is as follows:

$$\mu_{\text{RB}}^{\text{PPO}} = \frac{1}{M} \sum_{m=1}^{M} \frac{p_{\boldsymbol{\theta}}(\boldsymbol{Y}^{(m)})}{p_{\boldsymbol{\theta}_{\text{old}}}(\boldsymbol{Y}^{(m)})} \lim_{N \to \infty} \sum_{n=1}^{N} \underset{Y_n \sim p_{\boldsymbol{\theta}_{\text{old}}}}{\mathbb{E}} \left[ \log \frac{\vec{p}_{\boldsymbol{\theta}_{\text{old}}}(Y_n \mid \boldsymbol{Y}_{<n}^{(m)})}{\vec{q}(Y_n \mid \boldsymbol{Y}_{<n}^{(m)})} \right]. \tag{58}$$

Importantly, $\mu_{\text{RB}}^{\text{PPO}}$ does *not* give an unbiased estimate of $\mathbb{E}_{\boldsymbol{Y} \sim p_{\boldsymbol{\theta}}} \left[ \log \frac{p_{\boldsymbol{\theta}_{\text{old}}}(\boldsymbol{Y})}{q(\boldsymbol{Y})} \right]$, i.e.,

$$\mathbb{E}[\mu_{\text{RB}}^{\text{PPO}}] = \underset{\boldsymbol{Y} \sim p_{\boldsymbol{\theta}}}{\mathbb{E}} \left[ \lim_{N \to \infty} \sum_{n=1}^{N} \underset{\overline{Y}_n \sim p_{\boldsymbol{\theta}_{\text{old}}}}{\mathbb{E}} \left[ \log \frac{\vec{p}_{\boldsymbol{\theta}_{\text{old}}}(\overline{Y}_n \mid \overline{\boldsymbol{Y}}_{<n})}{\vec{q}(\overline{Y}_n \mid \overline{\boldsymbol{Y}}_{<n})} \right] \right] \neq \underset{\boldsymbol{Y} \sim p_{\boldsymbol{\theta}}}{\mathbb{E}} \left[ \log \frac{p_{\boldsymbol{\theta}_{\text{old}}}(\boldsymbol{Y})}{q(\boldsymbol{Y})} \right]. \tag{59}$$

Therefore, we caution the reader against using this estimator as a replacement for $\mu_{\text{MC}}^{\text{PPO}}$ in practice.

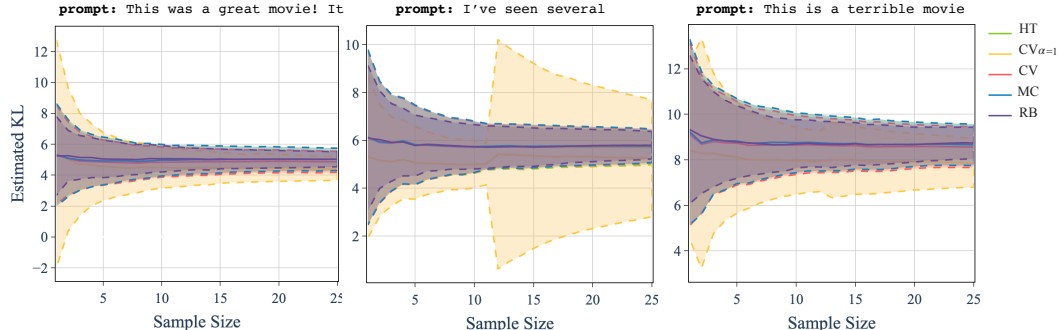

Figure 4: Comparing the bias, variance, and consistency of the estimators as the sample size increases. The $\mu_{\text{CV}}$ estimator with $\alpha = 1$ exhibits a higher standard deviation, particularly for neutral and negative prompts, where the variance becomes extremely large. In contrast, the RB estimator, $\mu_{\text{RB}}$, achieves the lowest standard deviation.

# E Additional Experiments

In Fig. 4, we visualize the KL estimates for three different prompts: (left) a positive prompt, (middle) a neutral prompt, and (right) a negative adversarial prompt. The traces represent the average estimates from all the repetitions, while the shaded regions indicate the standard deviation. Except $\mu_{\text{CV}}, \alpha = 1$, all other estimators are unbiased and consistent.

As the sample size increases, the chance of sampling from the tail of $p_{\boldsymbol{\theta}}$ also increases. These tail samples often correspond to negative movie reviews that had a high probability under the language model prior to fine-tuning, i.e., $q$, leading to extremely large values of $g(\boldsymbol{Y})$ and, consequently, a high standard deviation. This effect indeed depends on the prompt and is particularly pronounced for neutral and adversarial prompts.

We conducted an additional experiment to evaluate the estimators on a random subset of prompts from the UltraFeedback dataset [8]. We compute the KL divergence between `Zephyr-7B-Beta` [38] and its reference model, `Mistral-7B-v0.1` [18]. Zephyr is fine-tuned from Mistral using DPO on UltraFeedback, and as part of this fine-tuning, it is desirable not to diverge significantly from the base model.

We randomly sampled 512 prompts and generated 100 responses per prompt. For estimation, we used subsets of 1, 5, and 10 samples, reserving the remaining samples to estimate each method's standard deviation. Tab. 2 reports the KL estimate ± standard deviation for each estimator. Unlike our GPT-2 experiments, we had to use half-precision to perform inference and forward passes on a single GPU, which introduces bias in the HT and CV estimators. Consistent with our findings on the IMDB dataset, our proposed RB estimator consistently achieves the lowest standard deviation across all settings, reaffirming its stability and reliability.

Table 2: Estimated value ± empirical standard deviation of different estimators. RB estimator consistently achieves the lowest standard deviation.

|  | $M = 1$ | $M = 5$ | $M = 10$ |
|---|---|---|---|
| $\mu_{\text{MC}}$ | $18.05 \pm 3.19$ | $18.05 \pm 1.63$ | $18.05 \pm \mathbf{0.75}$ |
| $\mu_{\text{HT}}$ | $18.05 \pm 12.64$ | $18.56 \pm 5.34$ | $19.24 \pm 3.62$ |
| $\mu_{\text{CV1}}$ | $17.17 \pm 3.17$ | $17.17 \pm 1.62$ | $17.17 \pm 0.75$ |
| $\mu_{\text{CV}}$ | $17.80 \pm 3.18$ | $17.80 \pm 1.63$ | $17.80 \pm 0.75$ |
| $\mu_{\text{RB}}$ | $18.05 \pm \mathbf{3.16}$ | $18.05 \pm \mathbf{1.61}$ | $18.05 \pm \mathbf{0.75}$ |

# F Experimental Details

## F.1 Code Snippet

```python
def compute_kl(logprobs, ref_logprobs, logits, ref_logits):
    """
    Compute KL divergence using two estimators.

    Args:
        logprobs: Log probabilities of sampled actions from policy
        ref_logprobs: Log probabilities of same actions from reference
        logits: Full distribution logits from policy (all actions)
        ref_logits: Full distribution logits from reference (all actions)

    Example:
        # Policy samples action 3 from 1000 possible actions
        logprobs = [-2.3]          # Only action 3's log prob
        logits = [0.1, -0.5, ...]  # All 1000 action logits (raw)

        # MC: uses only sampled action
        # RB: uses all 1000 actions for exact expectation
    """
    # Monte Carlo: unbiased but higher variance
    kl_mc = mean(logprobs - ref_logprobs)

    # Rao-Blackwell: lower variance, uses full distribution
    log_p = log_softmax(logits)        # Normalize to log probs
    log_q = log_softmax(ref_logits)    # Normalize to log probs
    kl_rb = mean(sum(exp(log_p) * (log_p - log_q)))

    return kl_mc, kl_rb
```

## F.2 RLHF Experiments

In App. F.2, we include the hyperparameters used with the RLOO algorithm for the sentiment control experiment. Each experiment takes approximately 20 minutes on a single rtx_4090 GPU.

| Hypterparameter | Value |
|---|---|
| Optimizer | AdamW ($\epsilon = 1e-5$, $lr = 3e-6$) |
| Scheduler | Linear |
| Batch Size | 32 |
| $\beta$ | 0.07 |
| $k$ | 2 |
| Number of RLOO Updates Iteration Per Epoch | 4 |
| Clip range | 0.2 |
| Sampling Temperature | 1 |

