# OpenReview forum: "Better Estimation of the Kullback--Leibler Divergence Between Language Models"
_NeurIPS.cc/2025/Conference — NeurIPS 2025 poster_

### Official Review · Reviewer_oszg · 2025-06-30

**Clarity:** 3
**Significance:** 3
**Originality:** 3
**Rating:** 4
**Confidence:** 4

**Summary:**

This paper propose an estimator to the KL divergence and its gradient during RLHF, based on the decomposition of conditional expectation. The proposed estimation is shown to have lower variance, compared with the classicial Montecarlo exstimation, both in KL divergence itself and its gradient. The authors also use the proposed estimation to conduct RLHF and verify it induce a more stable training process.

**Questions:**

See Weankess, I will raise my score if the author can address my concerns in above.

**Ethical Concerns:**

["NO or VERY MINOR ethics concerns only"]

**Final Justification:**

See above

**Limitations:**

See Weakness.

**Quality:**

3

**Strengths And Weaknesses:**

Strengths:

This paper is well written and discuss an important question.

This paper has a strong theoretical guarantees of the proposed techniques.

Weakness:

My major concerns are two folds, respectively about the improvement over other varaince reduction techniques and the empirical evidance on the varaince reduction of gradient on KL divergence.

For the first, though the author compare their estimation with the vanilla MC method, but the improvement over the other variance reduction method is neglected. For example to comparison with  "Schulman. 2020. Approximating KL divergence."

For the second, the empirical results only verify the estimated KL divergence has lower variance, but the verification on the gradient of KL divergence is unexplored, which actually decides the dynamics of RLHF.

---

> ### Author Rebuttal · Authors · 2025-07-30
>
> We thank the reviewer for their thoughtful feedback.
>
> > For the first, though the author compare their estimation with the vanilla MC method, but the improvement over the other variance reduction method is neglected. For example to comparison with "Schulman. 2020. Approximating KL divergence."
>
> We want to clarify that the control variate estimator we discuss (CV with alpha = 1) is the same as the estimator proposed by Schulman (2020), known as the K3 estimator. We will make sure to emphasize this point further in the final version of the paper. In our experiments, we observe that this estimator (CV with alpha=1) can suffer from unbounded variance (figure 1), which limits its reliability compared to our proposed method.
>
> > For the second, the empirical results only verify the estimated KL divergence has lower variance, but the verification on the gradient of KL divergence is unexplored, which actually decides the dynamics of RLHF.
>
> In our RLHF experiments, we compare the variance of the MC and RB estimators at each training step. Figure 2 (right) shows the standard deviation of both estimators’ KL estimates throughout fine-tuning. We run RLHF five times with each estimator and plot the mean KL values along with their standard deviations (shown as shaded regions). The results clearly show that the MC estimator exhibits higher variance across training steps, while the RB estimator provides more stable and reliable KL estimates. Importantly, this improved stability translates to better downstream performance. As demonstrated in Figure 3, models trained using the RB estimator more consistently appear on the Pareto front balancing reward and KL divergence, highlighting the practical benefits of reduced variance in the training dynamics.

---

> > ### Comment · Reviewer_oszg · 2025-08-01
> > **Response**
> >
> > Thanks for applying, the author may misunderstand my second question. I said "but the verification on the gradient of KL divergence is unexplored" means a direct empirical verification on the varaince reduction of gradient similar to Figure 1. Though, a more stable learning dynamics can be somehow influenced by an accurate gradient estimator, but it also influenced by many other components, e.g., the setting of optimizer, the initialization points. Thus, in my opinion, a image similar to Figure 1 about the variance of gradient estimator can address my concern well.

---

> ### Author Response · Authors · 2025-08-03
> **Response to reviewer #oszg**
>
> Thank you very much for the clarification. We now better understand your concern and have conducted additional experiments to directly assess the variance of the gradient estimators for KL divergence.
>
> **Experimental Setup**
>
> To ensure a fair and controlled comparison, we directly measured the gradient variance as follows:
>
> - **Model**: GPT-2 fine-tuned on the IMDB sentiment classification task
> - **Evaluation Protocol**: For each estimator, we ran 5 independent trials, each using 5 batches of 8 prompts (generating 40 responses). All estimators used the same prompts and model initializations to ensure a fair comparison. In each trial, we computed gradients based on the model’s responses to the prompts, with new responses sampled in every run. We then measured the variance of the gradient norms across the 5 runs.
>
>
> **Results**
>
> | Estimator | Gradient Norm Variance | KL Estimate | Variance Ratio |
> |-----------|------------------------|------------|----------------|
> | **RB**    | 59.90                  | 7.81       | **0.75**       |
> | **MC**    | 79.44                  | 7.96       | –              |
>
> **Key Observations**
>
> 1. Unbiased KL Estimation: Both estimators yield comparable average KL values (7.81–7.96), confirming that they are unbiased.
>
> 2. Reduced Gradient Variance: The RB estimator demonstrates a **24.6% reduction in gradient variance** compared to the conventional MC estimator (variance ratio = 0.75), providing direct empirical evidence in line with the theoretical motivation.
>
> While we are unfortunately unable to include an additional figure at this stage due to NeurIPS formatting constraints, we will incorporate a detailed visualization and extended discussion in the final camera-ready version.
>
> We hope this addresses your concern, and we would be happy to clarify further if needed.

---

> > ### Comment · Reviewer_oszg · 2025-08-04
> >
> > Thanks for replying, this address my concerns, and I will raise my score.

---

### Official Review · Reviewer_A7gG · 2025-07-03

**Clarity:** 3
**Significance:** 3
**Originality:** 3
**Rating:** 4
**Confidence:** 3

**Summary:**

This paper addresses the problem of estimating the Kullback-Leibler (KL) divergence between language models, which is crucial for applications like reinforcement learning from human feedback (RLHF). The authors identify that while Monte Carlo (MC) estimation is commonly used for this task, it suffers from high variance and can even produce negative estimates of KL divergence, which is theoretically non-negative. To address these issues, the paper introduces a Rao-Blackwellized estimator that applies the classical variance reduction technique of Rao-Blackwellization to KL divergence estimation. The key insight is to decompose the language model estimation into step-wise components and compute exact conditional KL divergences at each step, which are then combined to form the final estimate. The authors prove that their estimator is unbiased and has variance less than or equal to the standard MC estimator, while maintaining the same computational complexity and guaranteeing non-negative estimates.

**Questions:**

Is there any assumption about the variance?

**Ethical Concerns:**

["NO or VERY MINOR ethics concerns only"]

**Limitations:**

See the weakness part.

**Paper Formatting Concerns:**

The reference formatting of reference is strange. e.g. "Kullback–Leibler [KL; 18]", "literature [8, 13, interalia]".

**Quality:**

3

**Strengths And Weaknesses:**

Strengths：

1. The paper provides rigorous theoretical analysis with formal proofs showing that the Rao-Blackwellized (RB) estimator is unbiased and has variance less than or equal to the Monte Carlo estimator.

2. A key strength is that the proposed method achieves variance reduction without additional computational overhead compared to the standard MC estimator.

3. The experiments systematically compare multiple estimators across different metrics (bias, variance, consistency) and demonstrate practical benefits in RLHF training, including improved stability and better Pareto frontier performance.

Weakness：

1. The core contribution applies an existing statistical technique (Rao-Blackwellization) to a specific domain. While the application is novel and valuable, the methodological innovation is somewhat incremental.

2. The empirical evaluation is restricted to relatively small models (GPT-2) and a single task (sentiment control). While this is acknowledged as a limitation due to computational constraints, it raises questions about scalability to modern large language models and diverse applications.

3. The theoretical results assume finite variance for both estimators, but the paper doesn't deeply analyze when these assumptions might be violated in practice or provide guidance for detecting such cases.

---

> ### Author Rebuttal · Authors · 2025-07-30
>
> We thank the reviewer for their thoughtful feedback.
>
> - Regarding the comment that our work applies an existing technique and is methodologically incremental, it is true that our method is based on Rao–Blackwellization, a well-known variance reduction technique. Our contribution lies in demonstrating that this simple, principled technique, requiring only minimal changes and reusing logits already computed during the forward pass, can substantially reduce variance at almost no additional cost. In fact, this simplicity could facilitate the integration of this method in RLHF libraries.
>
> - Regarding other model families and datasets, we refer the reviewer to the response [to the reviewer #Q7xD](https://openreview.net/forum?id=um9kHMof0c&noteId=WF9BPFMFcR).
>
>
> - Regarding the assumption of bounded variance, we thank the reviewer for raising this important point. We assume that the true KL divergence is finite. This assumption is generally realistic when p is a model fine-tuned to approximate q, often initialized from a checkpoint of q. However, in cases where q is close to zero while p remains positive, the KL divergence and consequently the variance of our estimators can become unbounded. We agree with the reviewer that this aspect deserves further clarification, and we will address it in the final version of the paper.

---

> > ### Comment · Reviewer_A7gG · 2025-08-07
> >
> > Thanks for the response with further explanation and additional results.  I would like to maintain my score.

---

### Official Review · Reviewer_5fbx · 2025-07-06

**Clarity:** 4
**Significance:** 2
**Originality:** 3
**Rating:** 4
**Confidence:** 4

**Summary:**

This paper introduces a Rao-Blackwellized estimator for KL divergence between language models, offering unbiased estimates with lower variance than standard Monte Carlo methods. It improves training stability in RLHF and yields models closer to the reward–KL Pareto frontier. To facilitate fine-tuning practices, this paper also derives a more stable and effective version of the estimator. The arguments are well-supported by both theoretical analysis and experimental results.

**Questions:**

Q1, It is out of the scope for this paper, but it would be much helpful if the dependency w.r.t. the reward function is clarified.

In many real-world datasets, the binary reward models, e.g., the Bradley-Terry(BT) model, have been known to be exposed to 'intransitivity' risk because it relies on scalar variables, where all preferences are transitive by assumption.
- The literature below studied representative preference datasets in the real world, where the 'transitive' relationship between preference annotations may not always hold.
- https://arxiv.org/abs/2409.19325 (Duan et al, 2017)

What is the implication or relationship between this 'intransitivity' and the proposed RB estimator?

**Ethical Concerns:**

["NO or VERY MINOR ethics concerns only"]

**Final Justification:**

The authors provided their candid response. The Pareto frontier result is impactful, and I will maintain my score as 'Weak Accept'.

**Limitations:**

yes

**Paper Formatting Concerns:**

No issue is identified regarding the submitted paper content, paper references, the NeurIPS paper checklist.

**Quality:**

4

**Strengths And Weaknesses:**

Strength:
1. This paper has proposed an RB estimator for KL divergence estimation between LLMs. Lower variance and unbiasedness are theoretically justified, despite the higher estimation cost.
2. The idea is straightforward, originating from the context of divergence estimation; a code snippet for implementing the RB estimator is provided.
3. Evaluation is mostly limited to sentiment control.

Weakness:
1. Broader alignment tasks and certain well-benchmarked datasets such as UltraFeedback remain to be validated.

---

> ### Author Rebuttal · Authors · 2025-07-30
>
> We thank the reviewer for their thoughtful feedback. For the comment regarding other model families and datasets, we refer the reviewer to the response [to the reviewer #Q7xD](https://openreview.net/forum?id=um9kHMof0c&noteId=WF9BPFMFcR).
>
> Questions:
> >  It is out of the scope for this paper, but it would be much helpful if the dependency w.r.t. the reward function is clarified. What is the implication or relationship between this 'intransitivity' and the proposed RB estimator?
>
> That is an interesting question. As discussed in the cited work, intransitivity in preference data can introduce significant challenges when learning a reliable reward model. In particular, it can lead to a reward function that is noisy or poorly calibrated. When optimizing with a KL-constrained RL objective in this setting, we suspect that the model may compensate for the weak reward signal by relying more heavily on the KL term, since it provides a more consistent optimization direction. However, if the KL estimator used in this process has high variance, this over-reliance can destabilize training. Our proposed RB estimator could help in this case by providing a more stable estimate of the KL divergence.

---

> > ### Comment · Reviewer_5fbx · 2025-08-03
> >
> > I thank the authors for their candid response. The pareto frontier result is impactful and I will maintain my score.

---

### Official Review · Reviewer_Q7xD · 2025-07-23

**Clarity:** 4
**Significance:** 3
**Originality:** 4
**Rating:** 5
**Confidence:** 3

**Summary:**

This paper proposes a new estimator of KL divergence between language models through Rao-Blackwellization. The proposed estimator is unbiased and has less variance than the baseline Monte Carlo estimator that is currently well used in machine learning applications. The paper introduces the Rao–Blackwellized Monte Carlo method and analyzes the gradient updates. The paper also presents experiments on preference finetuning with GPT-2 on the IMDB dataset. The experiments present bias and variances of different baseline estimators along with averaged rewards using different KL estimators. Results show the proposed RB estimator has low variance and achieves pareto fronts in evaluation.

**Questions:**

- I wonder if there is anyway to quantify the improvements in numerical stability. It could be a plus to show this in the experiment sections.
- Does the method apply to other divergences in the f-divergence family like Jensen–Shannon divergence?
- In the authors' opinion, what downstream tasks benefit most from lower variances of KL divergence estimation? It would be helpful to elaborate the potential impact of the proposed method.

**Ethical Concerns:**

["NO or VERY MINOR ethics concerns only"]

**Final Justification:**

Additional results strengthen the paper, but a higher score would need more extensive experiments from different real world applications.

**Limitations:**

Yes.

**Paper Formatting Concerns:**

No issues.

**Quality:**

3

**Strengths And Weaknesses:**

Strengths
- The paper gives detailed preliminaries and derivations of the proposed method such as Control Variate Monte Carlo and Rao–Blackwellization.
- The paper also analyzes gradient computation and also off-policy gradient for real-world settings.
- The paper implements and compares with a number of baselines thus making results convincing in experiments.

Weaknesses
- The experiments only include one preference finetuning task on one small-scale dataset IMDB-sentiment. The experiments are done with GPT-2 model. It would be more helpful to show the advantage of the proposed estimator in more real-world applications. These could be done with more diverse variety of experiments like experiments with other applications such as knowledge distillation which also use KL divergence a lot, or experiments with different tasks in preference finetuning domain, or experiments with different model families.

---

> ### Author Rebuttal · Authors · 2025-07-30
>
> We thank the reviewer for their thoughtful feedback.
>
> ## Experiments on another model family and real-world benchmarks
>
> We thank all the reviewers for acknowledging our limited computational resources to conduct large-scale experiments. That said, a recurring concern was the lack of evaluation on a different model family and a real-world benchmark (for example, reviewer `#5fbx` specifically suggested experiments on the UltraFeedback dataset)
>
> To address this, we conducted an additional experiment to evaluate all the discussed KL divergence estimators on a random subset of prompts from the UltraFeedback dataset. We compute the KL divergence between [Zephyr-7B-Beta](https://huggingface.co/HuggingFaceH4/zephyr-7b-beta) and its reference model, [Mistral-7B-v0.1](https://huggingface.co/mistralai/Mistral-7B-v0.1). Zephyr is fine-tuned from Mistral using DPO on UltraFeedback, and as part of this fine-tuning, it is desirable not to diverge significantly from the base model.
>
> We randomly sampled 512 prompts and generated 100 responses per prompt. For estimation, we used subsets of 1, 5, and 10 samples, reserving the remaining samples to estimate each method’s standard deviation. The table below reports the KL estimate ± standard deviation for each estimator. Unlike our GPT-2 experiments, we had to use half-precision to perform inference and forward passes on a single GPU, which introduces bias in the HT and CV estimators. Consistent with our findings on the IMDB dataset, our proposed RB estimator consistently achieves the lowest standard deviation across all settings, reaffirming its stability and reliability.
>
>
> | Method | M = 1 | M = 5 | M = 10 |
> |--------|--------|--------|---------|
> | HT     | 18.05 ± 12.64 | 18.56 ± 5.34 | 19.24 ± 3.62 |
> | CV1    | 17.17 ± 3.17 | 17.17 ± 1.62 | 17.17 ± 0.75 |
> | CV     | 17.80 ± 3.18 | 17.80 ± 1.63 | 17.80 ± 0.75 |
> | MC     | 18.05 ± 3.19 | 18.05 ± 1.63 | **18.05 ± 0.75** |
> | **RB** | **18.05 ± 3.16** | **18.05 ± 1.61** | **18.05 ± 0.75** |
>
> ## Questions
> > I wonder if there is anyway to quantify the improvements in numerical stability. It could be a plus to show this in the experiment sections.
>
> Thank you for this suggestion. The numerical instability arises when computing g(Y), and it's partially captured in Table 1 and Figure 1. However, it's more clearly illustrated in Appendix G (Figure 4). This issue typically occurs with samples from the tail of p (often negative movie reviews) that are still likely under q but suppressed during DPO, making p very small. As a result, the ratio q/p can overflow and bias the estimates. Figure 4 shows that this becomes more likely as the sample size increases (since tail samples are more likely to appear), especially for neutral or negative prompts. We will include a more detailed discussion in the final version of the paper.
>
> > Does the method apply to other divergences in the f-divergence family like Jensen–Shannon divergence?
>
> Yes. While our focus is on KL divergence due to its common use, the proposed estimator can also be applied to both forward and backward KL, which together can be used to estimate the Jensen–Shannon divergence.
>
> > In the authors' opinion, what downstream tasks benefit most from lower variances of KL divergence estimation? It would be helpful to elaborate the potential impact of the proposed method.
>
> This estimator is particularly useful when the number of samples available for estimating the divergence is limited, i.e., cases where standard Monte Carlo estimates of KL become highly unreliable. One important example is in RLHF, where GPU memory constraints the batch size, and therefore, the number of samples used for KL estimation is limited. In such settings, a lower-variance estimator can lead to more stable training and better-aligned models.

---

> > ### Comment · Reviewer_Q7xD · 2025-08-08
> >
> > Thank the authors for the additional experiments. These evidence from other models and dataset would strengthen the evaluation part of this work. I have read through the answers to my questions, which have addressed my concerns. I would keep my score.

---

### Decision · Program_Chairs · 2025-09-17

**Decision:**

Accept (poster)

**Comment:**

This paper proposes a Rao-Blackwellization of the typical sampling-based estimator of KL-divergence and its gradient, particularly in the context of its use in RLHF. They show that the new estimator has indeed smaller MSE theoretically and they also empirically demonstrate that this leads to better estimation in practice. The reviewers appreciate the rigor of the approach and its potential impact.

During the discussions, the chief concern was whether this impact was properly demonstrated to occur in practice. The authors provided additional experiments, which should be part of the paper. The results on assessing the gradient variance reduction and a discussion on the conditions necessary for the new estimator to be well behaved, should also be included. More RLHF experiments would better motivate the community to adopt this new estimator, considering that this context is the main driver of the research, and practitioners would heavily weigh whether the benefits of making this change are worthwhile. Additionally, the authors could explore other contexts where KL estimation is critical and where adopting the new estimator can make a significant difference.